# MD2 activation by direct AGE interaction drives inflammatory diabetic cardiomyopathy

Yi Wang[1,2,4✉], Wu Luo[2,4], Jibo Han[1,2,4], Zia A. Khan[2,3], Qilu Fang[2], Yiyi Jin[2], Xuemei Chen[2], Yali Zhang[2], Meihong Wang[2], Jianchang Qian[2], Weijian Huang[1], Hazel Lum[2], Gaojun Wu[1] & Guang Liang [iD] [1,2✉]

Hyperglycemia activates toll-like receptor 4 (TLR4) to induce inflammation in diabetic cardiomyopathy (DCM). However, the mechanisms of TLR4 activation remain unclear. Here we examine the role of myeloid differentiation 2 (MD2), a co-receptor of TLR4, in high glucose (HG)- and diabetes-induced inflammatory cardiomyopathy. We show increased MD2 in heart tissues of diabetic mice and serum of human diabetic subjects. MD2 deficiency in mice inhibits TLR4 pathway activation, which correlates with reduced myocardial remodeling and improved cardiac function. Mechanistically, we show that HG induces extracellular advanced glycation end products (AGEs), which bind directly to MD2, leading to formation of AGEs-MD2-TLR4 complex and initiation of pro-inflammatory pathways. We further detect elevated AGE-MD2 complexes in heart tissues and serum of diabetic mice and human subjects with DCM. In summary, we uncover a new mechanism of HG-induced inflammatory responses and myocardial injury, in which AGE products directly bind MD2 to drive inflammatory DCM.

---

[1] Department of Cardiology, the First Affiliated Hospital of Wenzhou Medical University, Wenzhou, Zhejiang 325035, China. [2] Chemical Biology Research Center, School of Pharmaceutical Sciences, Wenzhou Medical University, Wenzhou, Zhejiang 325035, China. [3] Department of Pathology and Laboratory Medicine, University of Western Ontario, London, Ontario N6A 5C1, Canada. [4] These authors contributed equally: Yi Wang, Wu Luo, Jibo Han. ✉email: yi.wang1122@wmu.edu.cn; wzmcliangguang@163.com

Diabetic cardiomyopathy (DCM) is a significant cause of morbidity and mortality in diabetic patients[1]. Multiple factors contribute to the development of DCM[1,2], with myocardial inflammation representing an early event[3–5]. Much evidence documents a causal relation between hyperglycemia and inflammation in diabetes, appearing to engage both direct and indirect regulation. High levels of glucose (HG) can directly promote inflammatory activities such as increased production of pro-inflammatory cytokines[6–8], in a wide range of cell types. However, the precise pathway(s) by which glucose initiates the pro-inflammatory cascades remain unclear.

Chronic inflammation in diabetes is believed to be linked to the activation of toll-like receptor 4 (TLR4), a component of the innate immune system[9]. TLR4 forms a cell-surface dimer with its co-receptor, myeloid differentiation factor 2 (MD2) to function as a recognition receptor for lipopolysaccharide (LPS). Upon LPS binding to MD2, LPS-MD2-TLR4 complex dimerizes and recruits myeloid differentiation primary response-88 (MyD88) and activates mitogen-activated protein kinases (MAPKs) and nuclear factor-kappa B (NFκB), which upregulate inflammatory cytokines[10]. In vitro studies show that HG activates TLR4 in myocytes[11], monocytes[12], endothelial cell[13], and kidney epithelial[13,14] and mesangial cells[15]. In addition, monocytes in human diabetic subjects show elevated levels of MD2 and TLR4[16]. Diabetic mice and rats also show increased TLR4 levels in heart tissues[17,18]. Predictably, inhibition of TLR4 confers significant protection against cardiac remodeling and dysfunction in diabetic animals[19–21]. It is interesting to note that TLR4 can be activated by multiple endogenous ligands present in the diabetic milieu[22]. Our previous study showed that saturated fatty acids are able to directly bind MD2 and activate the MD2-TLR4 signaling pathway[23]. MD2 also mediates angiotensin II-induced cardiac inflammation through direct binding to Ang II[24]. We have also reported that oxidized lipoproteins activate TLR4 through MD2 in atherosclerotic lesions[25]. Although the role of MD2 in hyperglycemia-induced cardiac inflammatory injury is unknown, these evidences point to the intriguing possibility that glucose or glucose-derived factors interact with MD2 to activate TLR4.

The goal of this study is to systematically investigate the precise mechanisms by which HG induces inflammatory activities in the context of DCM. Our findings indicate that HG triggers rapid generation of advanced glycation end products (AGEs), which directly bind MD2 and lead to the activation of MD2-TLR4 signaling complex and inflammatory cardiac tissue injury. We present a mechanism by which hyperglycemia triggers inflammatory injury in DCM, which underscores the importance of MD2 as a new potential key therapeutic target for DCM.

## Results

**MD2-TLR4 is activated in cardiac tissues of diabetic mice**. A mouse model of type 1 diabetes was developed by administering streptozotocin (STZ), as previously described by us[8] and others[26]. Analysis of cardiac tissues showed a fivefold increase in MD2 protein levels at 16 weeks following the onset of diabetes (Fig. 1a). TLR4 proteins were also increased at 16 weeks (Fig. 1a). These changes were associated with increased MD2-TLR4 complex formation as indicated by co-immunoprecipitation (Fig. 1b and Supplementary Fig. 1). To identify the cellular source of MD2 in heart tissues, we performed immunostaining of tissue slices. Our results indicate co-localized MD2 immunoreactivity with cardiomyocyte α-actin and macrophage F4/80 marker (Fig. 1c). In addition, some MD2 staining labeled cells devoid of α-actin and F4/80. The findings suggest that upregulated MD2 may primarily be derived from cardiomyocytes and infiltrated macrophages.

**MD2 deficiency reduces cardiac injury in diabetic mice**. We investigated the significance of the elevated MD2-TLR4 expression in the cardiac tissue of diabetic mice by administering STZ in $Md2^{-/-}$ (MD2KO) mice. Hyperglycemia was confirmed for up to 16 weeks in wild-type (WT-STZ) and $Md2^{-/-}$ diabetic mice (MD2KO-STZ) (Supplementary Fig. 2a). MD2KO did not show any significant changes in fasting blood glucose levels nor alter body weights (Supplementary Fig. 2a, b).

We stained heart tissues from mice with H&E (Fig. 2a) to measure cardiomyocyte cell size. Our results show increased cardiomyocyte size in WT-STZ mice but not MD2KO-STZ mice (Supplementary Fig. 2c). These results parallel serum creatine kinase-muscle/brain (CK-MB) levels, which were significantly elevated in WT-STZ mice and dampened in MD2KO-STZ (Table 1). WT-STZ mice also showed increased heart weight to body weight ratios indicating cardiac hypertrophy, which was reversed in MD2KO-STZ mice (Table 1). Moreover, MD2KO-STZ mice showed significantly less severe reduced ejection fraction and fraction shortening as evaluated by echocardiography (Table 1, Supplementary Fig. 3). Furthermore, E/A ratios and isovolumic relaxation time (IVRT) were increased in WT-STZ mice (Table 1), consistent with previous reports and indicative of diastolic dysfunction. These cardiac abnormalities in WT-STZ mice were associated with upregulation of genes (Fig. 2b) and proteins (Fig. 2c, Supplementary Fig. 4) important for tissue remodeling, such as atrial natriuretic peptide (ANP), collagen 1, matrix metalloproteinase-2 (MMP2), MMP9, and transforming growth factor β1 (TGFβ1). However, myocardial expression of these genes and proteins in MD2KO-STZ mice were comparable to MD2KO-Con (Fig. 2b, c, Supplementary Fig. 4). Furthermore, staining of heart tissues with Sirius Red and Masson's Trichrome indicated excessive fibrosis in WT-STZ mice but not MD2KO-STZ mice (Fig. 2d, e, Supplementary Fig. 5). These findings indicate that MD2 deficiency protects against morphological and functional cardiac abnormalities in diabetes.

We hypothesized that the protective effects of MD2 deficiency were attributed to blockade of inflammatory responses downstream of MD2-TLR4. Indeed, myocardial tissue of WT-STZ mice showed threefold increases in Tnfa and Il6 mRNA, however, these cytokines remained at control levels in MD2KO-STZ mice (Fig. 2f, g). NF-κB and MAPK, the major signaling targets of MD2-TLR4, were also activated in the myocardial tissue of WT-STZ mice, as indicated by increased inhibitor of κB (IκB) degradation and phosphorylated extracellular signal-regulated kinase (ERK), c-jun N-terminal kinase (JNK), and p38 (Fig. 2h, Supplementary Fig. 6). Analysis of heart tissues of MD2KO-STZ mice showed a lack of these activations.

We next assessed whether MD2 deficiency prevents infiltration of macrophages in the heart. We stained heart tissues for CD68 and show increased immunoreactivity in WT-STZ mice compared with WT-Con (Supplementary Fig. 7a). CD68 immunoreactivity was reduced in heart tissues of MD2KO-STZ mice, suggesting decreased infiltration. Immunoblotting also showed increased CD68 protein levels in heart lysates prepared from WT-STZ mice but not MD2KO-STZ mice (Supplementary Fig. 7b). Furthermore, MD2KO-STZ mice showed no induction of inflammatory cell adhesion molecules ICAM1 and VCAM1 (Supplementary Fig. 7b). These findings suggest that MD2 deficiency reduces adhesion molecules involved in macrophage recruitment and prevents macrophage infiltration in diabetes.

Compound L6H21 is a selective MD2 inhibitor with anti-inflammatory activities[27]. We investigated whether pharmacological MD2 inhibition by L6H21 also protects against the development of myocardial injury in diabetic mice. As noted for MD2KO mice, L6H21 treatment did not alter STZ-induced decreased body weights or fasting blood glucose levels

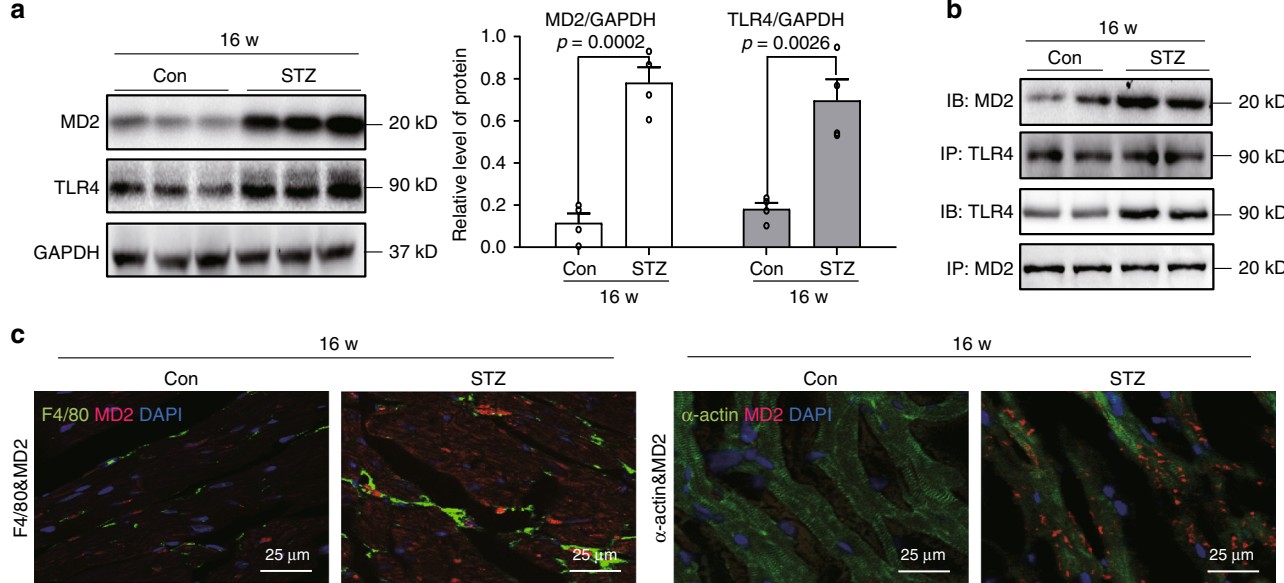

**Fig. 1 MD2-TLR4 complex activation in hearts of diabetic mice.** A mouse model of type 1 diabetes mellitus was developed by administering streptozotocin to C57BL/6 mice. Heart tissues were harvested at 16 weeks [Con = non-diabetic controls, STZ = diabetic mice]. **a** Representative immunoblot for MD2 and TLR4 in mouse cardiac tissue. GAPDH was used as loading control. Densitometric quantification of blots showing MD2 (white bars) and TLR4 (black bars) [$n = 4$; 3 Con and 3 STZ samples shown in immunoblots; means ± SEM]. **b** Representative immunoblots showing co-immunoprecipitation of TLR4 and MD2 in mouse heart tissues at 16 weeks following onset of diabetes [IP = precipitating antibody, IB = immunoblot antibody; $n = 4$; 2 Con and 2 STZ samples shown in immunoblots]. **c** Representative immunofluorescence staining of mouse heart tissues at 16 weeks for MD2 (red), macrophage marker F4/80 (green), and myocyte marker α-actin (green). Slides were counterstained with DAPI (blue) [$n = 4$]. Source data are provided as a Source Data file.

(Supplementary Fig. 8a, b). However, L6H21 treatment did prevent STZ-induced fibrosis (Supplementary Fig. 8c), as well as tissue remodeling genes (Supplementary Fig. 8d, e). Moreover, STZ-induced increases in myocardial TNF-α mRNA and protein (Supplementary Fig. 8f–h), Il6 (Supplementary Fig. 8i), as well as IκB degradation (Supplementary Fig. 8g) were significantly reduced by L6H21. These results show that L6H21 protects against diabetes-induced inflammatory heart injuries, similar to genetic MD2 deficiency.

**MD2-TLR4 in cardiac cells induces inflammatory responses.** Our next objective was to determine whether a specific cell type in the cardiac tissue is important for MD2-TLR4 signaling. Our results indicated MD2 co-localization with α-actin and F4/80 markers (Fig. 1c). However, cells that were not marked by α-actin and F4/80 also showed slight MD2 immunoreactivity. In addition to cardiomyocytes and macrophages, cardiac tissue contains abundant fibroblasts and endothelial cells. Therefore, we harvested and compared the four cell types for MD2 expression. Our results show high levels of MD2 mRNA and protein in primary cardiomyocytes and macrophages (Supplementary Fig. 9). Although endothelial cells and fibroblasts express MD2, the levels were significantly lower compared with macrophages and cardiomyocytes.

To build on these results, we reconstituted marrows of irradiated MD2KO mice with marrow cells harvested from WT mice or MD2KO mice. We then made these mice diabetic by STZ and examined their heart tissues after 12 weeks. Compared with sham WT mice, reconstitution of MD2KO mice with WT marrow cells was associated with reduced fibrosis (Supplementary Fig. 10a) and lower mRNA levels of Anp, Col1a1, Mmp2, and Tgfb1 (Supplementary Fig. 10b). These results show the importance of cardiac MD2 in unregulated STZ-induced tissue remodeling. Analysis of heart tissues from MD2KO mice

reconstituted with MD2KO marrow cells showed further reductions in mRNA levels of Anp, Col1a1, and Mmp2, as expected. Together with MD2 protein levels in harvested cardiac cells, our results point to cardiomyocyte and macrophage as primary cellular targets of MD2-TLR4 signaling in diabetes.

**Glucose activates MD2-TLR4 in cardiomyocytes and macrophages.** We next examined how HG promoted inflammatory responses through MD2-TLR4 signaling. As mentioned above, numerous studies have indicated increased levels of TLR4 pathway proteins in diabetes and cells exposed to HG[11–15]. Inhibition of TLR4 also reduces inflammatory cytokine expression, implicating activated TLR4 signaling upon HG exposure of cells[13,28]. To understand how HG activates TLR4, we first excluded osmotic stress contributing to inflammatory signaling. Rat heart-derived H9C2 cells and mouse primary peritoneal macrophages (MPMs) showed HG-induced TNF-α and IL-6 production (Supplementary Fig. 11a–e). These inductions were not seen with mannitol indicating the involvement of inflammatory signaling and not osmotic stress. We next determined the effects of HG in MD2-TLR4 complex formation in H9C2 cells. We exposed H9C2 cells to HG (33 mM glucose) for up to 30 min and analyzed MD2-TLR4 complex formation by co-immunoprecipitation. We found that HG induced MD2-TLR4 complex by 5 min, and this interaction was sustained for 30 min (Fig. 3a). Using glucose concentration ranging from 7 to 60 mM, we show significant MD2-TLR4 complex formation occurring at levels ≥30 mM (Fig. 3b). Moreover, H9C2 cells co-transfected with Flag-TLR4 and HA-TLR4 plasmids indicated that HG significantly induces TLR4 dimerization at 15 min (Supplementary Fig. 12). These findings support the thesis that HG induces rapid MD2-TLR4 complex formation and TLR4 dimerization. Pretreatment of H9C2 cells with the MD2 inhibitor L6H21 prior to HG exposure significantly

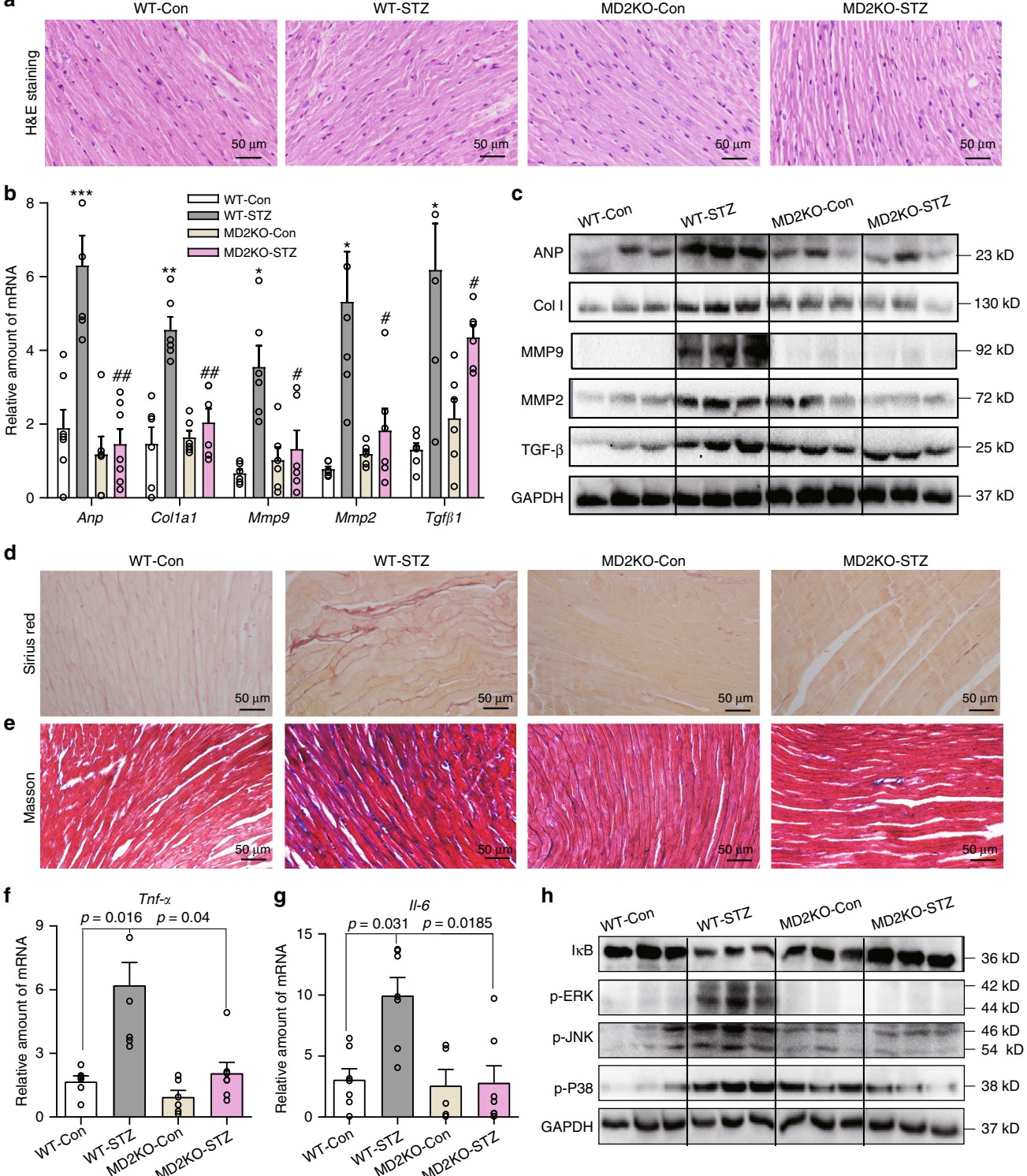

**Fig. 2 MD2 deficiency prevents diabetes-induced cardiac injury.** Diabetes was induced in C57BL/6 wild-type and MD2$^{-/-}$ mice by streptozotocin. Heart tissues were harvested at 16 weeks [WT-Con = non-diabetic wild-type controls, WT-STZ = diabetic wild type, MD2KO-Con = non-diabetic MD2$^{-/-}$, MD2KO-STZ = diabetic MD2$^{-/-}$]. **a** Representative H&E staining of cardiac tissues [$n = 6$]. **b** mRNA levels of cardiac tissue *Anp*, *Col1a1*, *Mmp9*, *Mmp2*, and *Tgfb1* normalized to *Actb* [means ± SEM; $n = 6$ per group; *$p < 0.05$, **$p < 0.01$, ***$p < 0.001$ compared with WT-Con; #$p < 0.05$, ##$p < 0.01$ compared with WT-STZ; *P*-values by unpaired *t* test are indicated]. **c** Representative immunoblots showing cardiac tissue ANP, Col 1, MMP-9, MMP-2, and TGF-β. GAPDH was used as a loading control [$n = 6$; 3 samples per group shown]. **d, e** Representative staining images of mouse heart sections showing Sirius Red (**d**) and Masson's Trichrome (**e**) ($n = 6$ per group). **f, g** qPCR analysis of *Tnfa* and *Il6* mRNA levels in cardiac tissues [means ± SEM; $n = 6$ per group]. **h** Representative immunoblots showing levels of IκB and phosphorylated ERK, JNK, and P38 in mouse cardiac tissues. GAPDH was used as loading control [$n = 6$; 3 samples per group shown]. Source data are provided as a Source Data file. *P*-values by one-way ANOVA in **b**, **f**, and **g** followed by Tukey's post hoc test are indicated.

**Table 1 Biometric and echocardiographic parameters of DM1 mice.**

| Parameter | WT-Con (n = 6)[a] | WT-STZ (n = 7)[a] | MD2KO-Con (n = 6)[a] | MD2KO-STZ (n = 7)[a] |
|---|---|---|---|---|
| IVSd (mm) | 0.84 ± 0.03 | 0.74 ± 0.02* | 0.77 ± 0.01 | 0.77 ± 0.01# |
| IVSs (mm) | 1.10 ± 0.08 | 0.95 ± 0.02* | 0.96 ± 0.02 | 0.96 ± 0.01# |
| LVd (cm) | 0.26 ± 0.01 | 0.24 ± 0.02 | 0.25 ± 0.01 | 0.25 ± 0.01 |
| LVs (cm) | 0.14 ± 0.003 | 0.13 ± 0.002 | 0.14 ± 0.007 | 0.14 ± 0.005 |
| FWd (mm) | 0.88 ± 0.037 | 0.73 ± 0.009*** | 0.77 ± 0.009 | 0.76 ± 0.012# |
| FWs (mm) | 1.123 ± 0.07 | 0.96 ± 0.02** | 0.99 ± 0.01 | 0.98 ± 0.02# |
| PWd (mm) | 0.84 ± 0.03 | 0.75 ± 0.02* | 0.76 ± 0.02 | 0.78 ± 0.01# |
| PWs (mm) | 1.06 ± 0.04 | 0.95 ± 0.02 | 0.98 ± 0.02 | 0.98 ± 0.02 |
| EF% | 82.12 ± 2.0 | 70.57 ± 2.3** | 80.78 ± 1.7 | 80.37 ± 1.3## |
| FS% | 44.48 ± 1.9 | 36.14 ± 2.0** | 43.50 ± 1.7 | 42.84 ± 1.2## |
| E (m/s) | 0.452 ± 0.07 | 0.458 ± 0.06 | 0.612 ± 0.04 | 0.583 ± 0.05 |
| A (m/s) | 0.531 ± 0.02 | 0.322 ± 0.06 | 0.632 ± 0.06 | 0.628 ± 0.08 |
| E/A | 0.842 ± 0.13 | 1.615 ± 0.20** | 1.002 ± 0.09 | 1.033 ± 0.15# |
| IVRT(ms) | 12.1 ± 1.38 | 21.14 ± 1.86* | 13.8 ± 2.12 | 16.3 ± 2.22# |
| HW (mg) | 125.1 ± 3.28 | 117.8 ± 1.42 | 131.0 ± 4.55 | 116.0 ± 2.92 |
| BW (g) | 29.68 ± 0.95 | 25.30 ± 0.38 | 32.17 ± 1.70 | 27.07 ± 0.48 |
| HW/BW (mg/g) | 4.218 ± 0.065 | 4.660 ± 0.031** | 4.093 ± 0.082 | 4.281 ± 0.075# |
| CK-MB | 22.15 ± 4.174 | 59.47 ± 4.200*** | 28.88 ± 1.835 | 37.25 ± 2.820## |

*IVSd* interventricular septal dimension in diastole, *IVSs* interventricular septal dimension in systole, *LVd* left ventricle in diastole, *LVs* left ventricle in systole, *FWd* left ventricular free wall in diastole, *FWs* left ventricular free wall in systole, *PWd* posterior wall thickness in diastole, *PWs* posterior wall dimension in systole, *EF* ejection fraction, *FS* fractional shortening, *E* peak mitral E velocity, *A* peak mitral A velocity, *IVRT* isovolumic relaxation time, *HW* heart weight, *BW* body weight, *CK-MB* creatine kinase-muscle/brain.
[a]Values shown as mean ± SEM.
*$p < 0.05$, **$p < 0.01$, ***$p < 0.001$ compared with WT-Con; #$p < 0.05$, ##$p < 0.01$ compared with WT-STZ. P-values by one-way ANOVA followed by Tukey's post hoc test are indicated.

blocked MD2-TLR4 complex formation (Fig. 3c), suggesting that MD2 was essential in mediating HG-induced TLR4 activation.

Activation of the MD2-TLR4 complex leads to recruitment of MyD88, resulting in activation of MAPK and NF-κB. We silenced MD2 using targeted siRNAs in H9C2 cells (Fig. 3d). In negative control-transfected H9C2 cells, HG increased immunoprecipitation pull-down of MyD88 with TLR4, but not in cells with MD2 knockdown (Fig. 3e). Moreover, MD2 knockdown in H9C2 cells reduced HG-induced degradation of IκB (Fig. 3f), indicative of NF-κB inhibition, and inhibited HG-induced phosphorylation of MAPK (ERK, JNK, and P38) (Fig. 3g). These findings were corroborated in cells treated with MD2 inhibitor L6H21 and MD2 neutralizing antibody (Supplementary Fig. 13a). Not surprisingly, both MD2 knockdown (Fig. 3h, i) and L6H21 (Supplementary Fig. 13b, c) significantly inhibited HG-induced mRNA of *Tnfa* and *Il6*.

To build on these results, we evaluated the consequence of TLR4 deficiency in HG-induced inflammatory signaling. siRNA-mediated knockdown of TLR4 reduced *Tlr4* levels to ~50% of control cells (Supplementary Fig. 14a) and effectively prevented HG-induced increases in *Tnfa* and *Il6* (Supplementary Fig. 14b, c).

We confirmed these findings from H9C2 in primary cardiomyocytes from Sprague-Dawley rats. Our results indicate that L6H21 pretreatment at 10 μM inhibits HG-induced MD2-TLR4 complex formation, IκB degradation, MAPK activation, and *Tnfa* levels in rat primary cardiomyocytes (Supplementary Fig. 15). We then examined macrophages and show that L6H21 prevents HG-induced formation of MD2-TLR4 complex and TLR4-MyD88 complex in MPMs (Fig. 3j). L6H21 pretreatment also inhibited HG-induced TNF-α and IL-6 production by MPMs (Fig. 3k, l). Induction of TNF-α and IL-6 by HG was not seen in MPMs harvested from MD2KO mice, confirming that HG activates MD2-dependent pro-inflammatory signaling.

Our in vivo data indicated that MD2 deficiency prevents cardiac fibrosis. Therefore, we evaluated indices of tissue remodeling in primary cardiomyocytes exposed to HG. We show that HG exposure for 12–24 h increases *Tgfb1* and *Mmp2*, however, siRNA-mediated MD2 knockdown prevented these HG-induced increases (Supplementary Fig. 16a). Moreover,

L6H21 pretreatment of rat primary cardiomyocytes inhibited HG-induced increases in protein levels of TGFβ, MMP-2, Collagen 1, and myosin heavy chain (MyHC) (Supplementary Fig. 16b, c). These in vitro findings corroborate our data obtained from STZ-induced model of type 1 diabetes, supporting a MD2-dependent mechanism.

**HG likely activates MD2-TLR4 indirectly.** Our next goal was to explore the mechanisms by which HG activates MD2. Our in vitro findings indicated that HG rapidly stimulates the formation of MD2-TLR4 complex in H9C2 cells and MPMs (Fig. 3a). MD2 is an extracellular protein, and functions as a co-receptor of TLR4 at the cell surface. As shown in Fig. 4a, there are two potential extracellular pathways by which HG may activate MD2-TLR4 complex: (1) HG directly interacts with MD2 protein or (2) HG works in concert with extracellular factors (e.g., in blood and interstitium) to interact with MD2.

We first examined if the glucose uptake and intracellular glucose is required for rapid MD2-TLR4 activation. Cardiac myocytes express glucose transporter-4 (GLUT4) to transport glucose into cells[29]. We found that increasing extracellular glucose levels by GLUT4 knockdown in H9C2 cells (Fig. 4b) results in enhanced HG-induced MD2-TLR4 complex formation (Fig. 4c). Excessive glucose in cells increased intracellular reactive oxygen species (ROS), which has been reported to induce TLR4[12]. We tested the idea that HG-induced ROS may modulate TLR4 in macrophages and H9C2 cells. Our results showed that ROS scavenger N-acetylcysteine (NAC) did not reduce the MD2-TLR4 interaction in cells stimulated with HG for 15 min (Supplementary Fig. 17a, b). As reported previously[12], pretreatment of cells with NAC reduced the levels of *Tnfa*, *Il6*, and *Tlr4* induced by a 12-h HG challenge (Supplementary Fig. 17c, e). These results show that intracellular HG increases the expression of *Tlr4* via ROS but is not involved in MD2-TLR4 complex formation and TLR4 activation.

We then explored the two potential mechanisms by which extracellular glucose may activate MD2-TLR4 complex at the cell surface. We tested whether glucose directly binds to the MD2 protein, like LPS. The direct glucose-MD2 interactions were

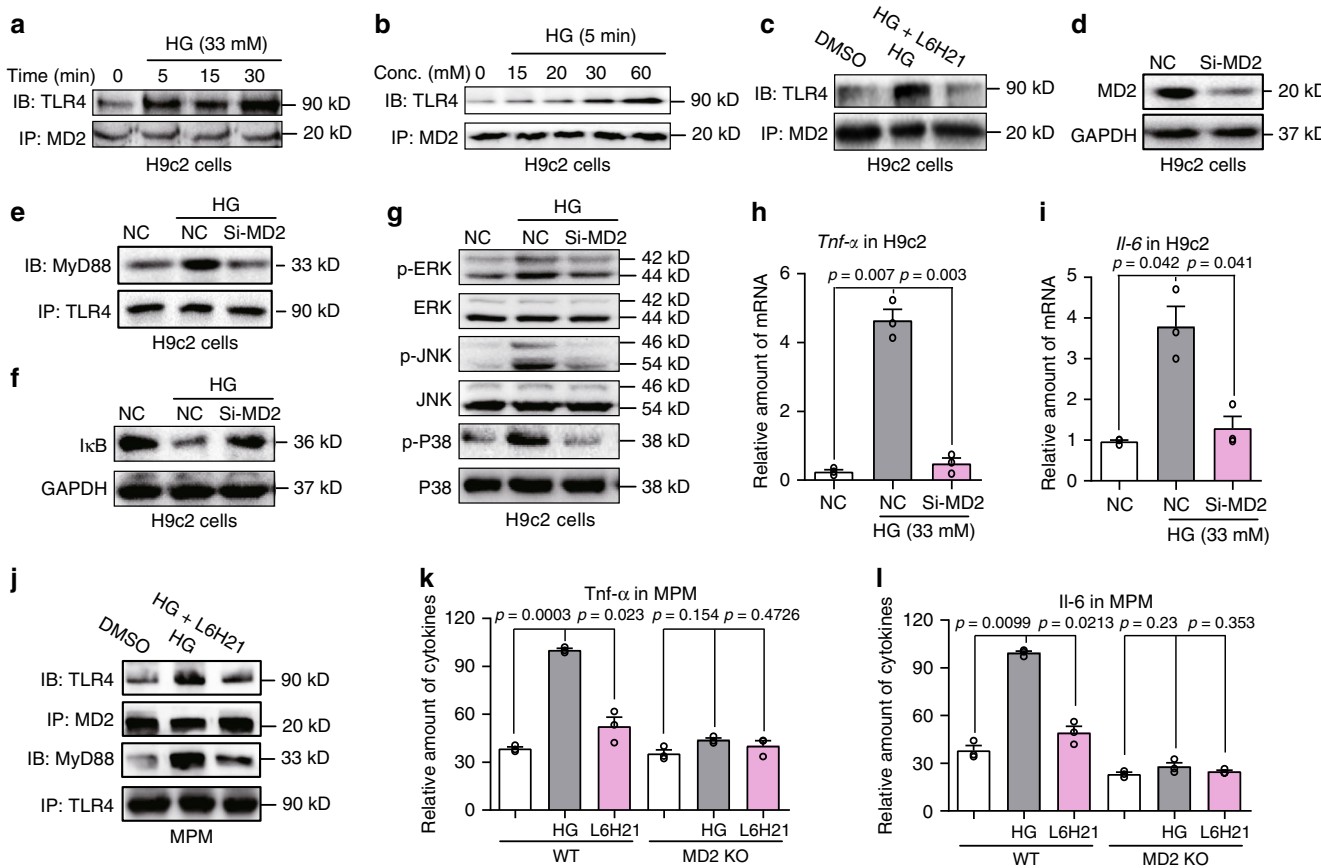

**Fig. 3 High glucose activates MD2-TLR4 signaling in cardiac cells. a**, **b** Representative immunoblots showing co-immunoprecipitation of MD2 and TLR4 in H9C2 cells exposed to HG for varying time points (**a**) and concentration (**b**) [n = 3]. **c** H9C2 cells were pretreated with 10 μM MD2 inhibitor L6H21 for 30 min before exposure to HG (33 mM glucose) for 5 min. Representative immunoblots showing co-immunoprecipitation of MD2 and TLR4 [n = 3]. **d** Western blot analysis showing levels of MD2 protein following transfection of H9C2 cells with MD2 siRNA [si-MD2 = MD2 targeting siRNA, NC = negative control; n = 3]. **e** Co-immunoprecipitation of TLR4 and MyD88 in H9C2 cells transfected with MD2 siRNA (si-MD2) and exposed to HG (33 mM glucose, 15 min) [n = 3]. **f**, **g** Representative blots of IκB and phosphorylation of ERK, JNK, and P38 in H9C2 cells transfected with MD2 siRNA and exposed to HG (33 mM glucose, 15 min). GAPDH and total MAPK proteins served as controls [n = 3]. **h**, **i** *Tnfa* and *Il6* mRNA levels in H9C2 cells transfected with MD2 siRNA (si-MD2) and challenged with HG (33 mM glucose, 6 h) [means ± SEM; n = 3 independent examinations]. **j** Co-immunoprecipitation of TLR4, MyD88, and MD2 in mouse peritoneal macrophages (MPMs). MPMs were pretreated with 10 μM MD2 inhibitor L6H21 for 30 min before exposure to HG (33 mM glucose). MD2-TLR4 complex was assessed following 5 min of HG exposure and TLR4-MyD88 complex at 15 min of HG exposure [n = 3 examinations]. **k**, **l** Levels of TNF-α and IL-6 in culture media of MPMs isolated from non-diabetic WT and MD2KO mice. Cells were pretreated with 10 μM L6H21 for 1 h and then exposed to HG (33 mM glucose) for 24 h. TNF-α and IL-6 levels were determined by ELISA [means ± SEM; n = 3 examinations]. Source data are provided as a Source Data file. *P*-values by one-way ANOVA in **h**, **i**, **k**, and **l** followed by Tukey's post hoc test are indicated.

determined using recombinant human MD2 (rhMD2) in an isothermal titration calorimetry (ITC) assay. Surprisingly, the analysis was negative, indicating no direct glucose-rhMD2 interaction (Fig. 4d). Therefore, we evaluated the importance of serum in HG-induced inflammatory responses (alternate extracellular mechanism). Results indicated that HG-induced (33 mM glucose, 24 h) TNF-α and IL-6 in MPMs was suppressed in the absence of serum (Fig. 4e). These results contrast with LPS (0.5 μg/mL, 24 h) which induces TNF-α and IL-6 in the presence or absence of serum (Fig. 4f). Similar responses were observed in H9C2 cells using qPCR analysis of cytokine mRNA (Supplementary Fig. 18). Interestingly, addition of 5 or 10% fetal bovine serum (FBS) to MPMs which were originally cultured without serum, restores HG-responsiveness and induces TNF-α and IL-6 (Fig. 4g, h). HG also failed to induce the formation of MD2-TLR4 and TLR4-MyD88 complexes in the absence of serum, whereas addition of serum rescues complex formation and recruitment of MyD88 (Fig. 4i). Similar responses to serum occurred in H9C2 cells, in which HG was unable to induce cytokine production, MD2-TLR4 complex formation, or recruit MyD88 (Fig. 4j–l).

The requirement of serum in HG-responsiveness of MPMs and H9C2 cells raises several important questions. Although we cultured and performed our studies in heat-inactivated serum, complement activation may potentially be involved HG-induced TNF-α and IL-6 production. To test this, H9C2 cells were cultured in media containing 10% heat-inactivated FBS or serum that did not undergo heat-inactivation, for 10 days. Cells were then exposed to HG (33 mM glucose) in the respective serum groups for 24 h and mRNA levels of *Tnfa* and *Il6* were determined. We show no significant difference in HG-induced *Tnfa* and *Il6* (Supplementary Fig. 19). We also tested whether altered glycolysis was responsible for MD2-TLR4 interaction in cells exposed to HG. We assessed glycolytic function through extracellular acidification rate (ECAR) in H9C2 and MPMs exposed to HG. Our results indicate increased glycolysis in MPMs but not H9C2 cells exposed to 33 mM glucose (Supplementary Fig. 20). To rule out glycolysis in inducing MD2-TLR4 activation, we pretreated H9C2 cells with glycolysis inhibitor 2-deoxy-glucose (2-DG) and then exposed the cells to HG. Co-immunoprecipitation showed that 2-DG does not alter HG-

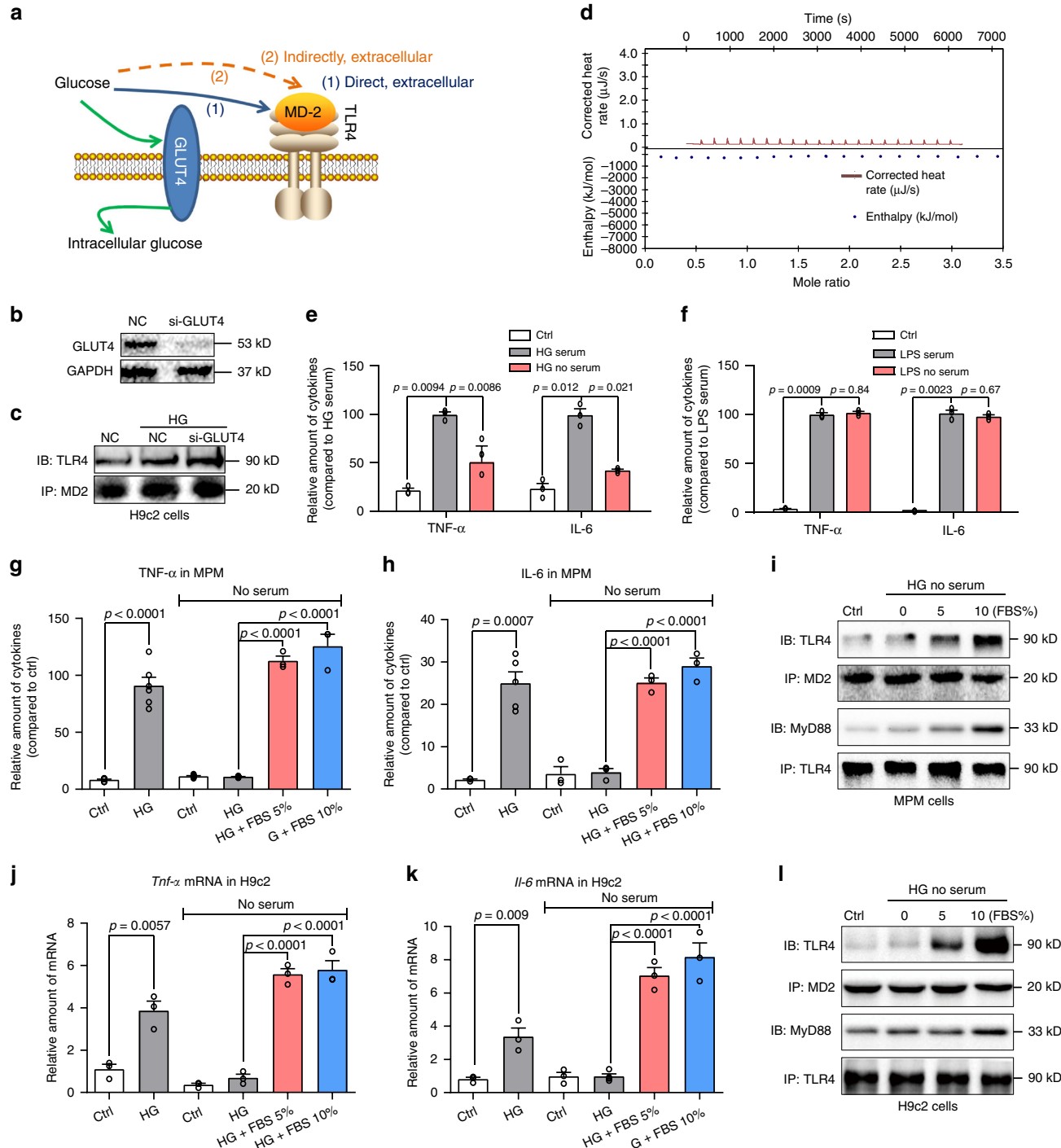

**Fig. 4 HG-mediated MD2-TLR4 activation requires serum. a** Schematic illustrating the potential modes of HG activating MD2-TLR4. **b** Immunoblot showing GLUT4 levels in H9C2 cells following siRNA transfection [si-GLUT4 = GLUT4 siRNA, NC = negative control; n = 3]. **c** TLR4-MD2 complex in H9C2 cells transfected with si-GLUT4 and exposed to HG for 5 min [n = 3]. **d** Isothermal titration calorimetry to detect glucose and human recombinant MD2 protein interaction (three independent experiments). **e, f** MPMs were exposed to HG in the presence or absence of serum for 24 h (**e**). Similarly, MPMs were exposed to LPS (**f**). Levels of TNF-α and IL-6 in culture media were determined [Ctrl = control with 10% FBS, HG/LPS serum = 33 mM glucose or 0.5 μg/mL LPS with 10% FBS, HG/LPS no-serum = 33 mM glucose or 0.5 μg/mL LPS with no-serum; means ± SEM; n = 3 examinations]. **g, h** MPMs were exposed to HG in the presence or absence of FBS for 24 h. MPMs in serum (first two bars on left) were expanded in media containing 10% FBS and exposed to HG in media containing 10% FBS. MPMs in no-serum (four bars on right) were expanded in media containing 10% FBS, serum-starved for 24 h, and the exposed to HG with indicated levels of FBS. Levels of TNF-α (**g**) and IL-6 (**h**) were determined in culture medium [means ± SEM; n = 3 examinations]. **i** Immunoblot showing TLR4-MyD88 and MD2-TLR4 complexes in MPMs exposed to HG in media containing FBS [Ctrl=media without serum; n = 3 independent experiments]. **j, k** H9C2 cells were exposed to HG in the presence or absence of FBS for 24 h. H9C2 cell treatments were carried out as described for MPMs. Levels of TNF-α (**i**) and IL-6 (**k**) were determined [means ± SEM; n = 3 examinations]. **l** Immunoblot showing TLR4-MyD88 and MD2-TLR4 complexes in H9c2 cells exposed to HG in media containing FBS [Ctrl=media without serum; n = 3]. Source data are provided as a Source Data file. *P*-values by one-way ANOVA in **e**, **f**, **g**, **h**, **j**, and **k** followed by Tukey's post hoc test are indicated.

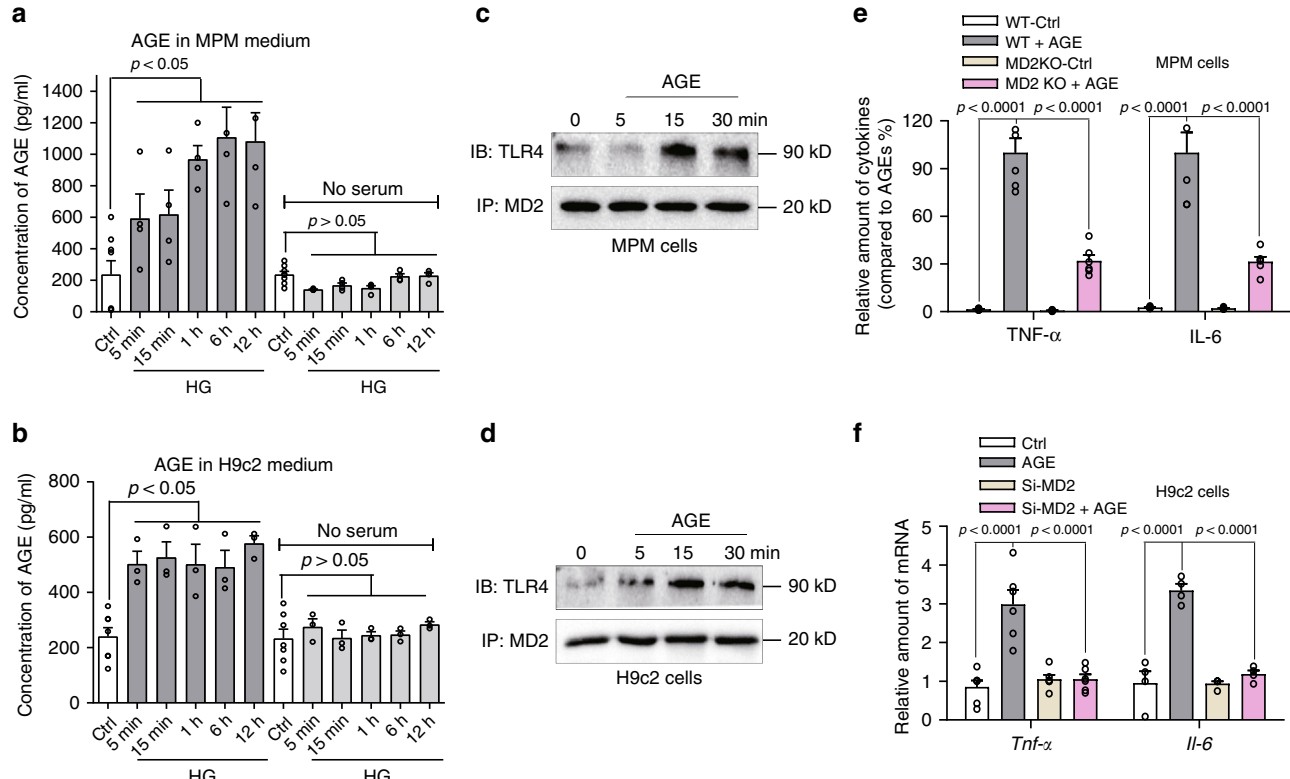

**Fig. 5 AGE products stimulate MD2-dependent inflammatory responses. a** AGE product formation in MPMs exposed to HG in the presence or absence of serum. MPMs were exposed to 33 mM glucose for different time periods in media containing 0 or 10% FBS. Levels of AGE products were determined in conditioned medium by ELISA [means ± SEM; $n = 4$ examinations]. **b** AGE product formation in H9C2 cells exposed to HG in the presence or absence of serum. H9C2 cells were exposed to 33 mM glucose for different time periods in media containing 0 or 10% FBS. Levels of AGE products were determined in conditioned medium by ELISA [means ± SEM; $n = 4$ examinations]. **c** Representative immunoblot showing co-immunoprecipitation of MD2-TLR4 complex in MPMs exposed to 33 µg/mL AGE-BSA [$n = 6$]. **d** Representative immunoblot showing co-immunoprecipitation of MD2-TLR4 complex in H9C2 cells exposed to 33 µg/mL AGE-BSA [$n = 3$]. **e** Levels of TNF-α and IL-6 in condition media of MPMs exposed to 33 µg/mL AGE-BSA for 24 h. MPMs isolated from WT or MD2KO mice were tested [means ± SEM; $n = 4$ examinations]. **f** Levels of *Tnfa* and *Il6* mRNA in H9C2 cells exposed to 33 µg/mL AGE-BSA for 6 h. H9C2 cells were transfected with control siRNA or siRNA targeting MD2 (siMD2) before treatments [means ± SEM; $n = 6$ examinations]. Source data are provided as a Source Data file. *P*-values by one-way ANOVA in **a**, **b**, **e**, **f** followed by Tukey's post hoc test are indicated.

induced MD2-TLR4 interaction in H9C2 cells (Supplementary Fig. 21a). Similarly, 2-DG dose not suppress HG-induced *Tnfa* and *Il6* in H9C2 cells (Supplementary Fig. 21b, c). Taken together, our results show that HG activates MD2-TLR4 signaling and this response is dependent on the presence of serum.

**HG produces AGE products that activate MD2.** AGE product generation is a well-established event for various proteins exposed to HG, and the pathogenesis of diabetes and cardiovascular diseases. We speculated that HG may produce AGE products in serum-containing medium to initiate MD2-dependent inflammatory responses. We exposed MPMs and H9C2 cells to HG in the presence or absence of serum for increasing duration and assessed AGE formation. Our results show that MPMs exposed to HG in the absence of serum were unable to generate AGEs; whereas in the presence of serum, HG increased AGE products by approximately threefold within 5 min (Fig. 5a). Similarly, HG increased AGE products in H9C2 cell media when exposed in the presence of serum (Fig. 5b).

Since AGE products are heterogenous, formed from different intermediates, and have differential activities in cells[30], we measured methylglyoxal (MGO) and Nε-carboxymethyl-lysine (CML) in media of H9C2 exposed to HG. Our results show that HG exposure of H9C2 cells for up to 15 min does not increase

CML levels and no changes were seen in MGO levels for up to 60 min HG exposure (Supplementary Fig. 22a–c). To confirm these extracellular reactions, we incubated cell-free complete growth media containing serum with 33 mM glucose for 15 min and measured the levels of AGE-, MGO-, and CML-modified proteins. Our results show that glucose exposure of serum-containing media rapidly generates AGE-modified serum proteins, while we did not observe increases in MGO or CML (Supplementary Fig. 22d).

At least in our system, at appears that AGE-modified proteins may be involved in HG inflammatory responses. Direct exposure of MPMs and H9C2 cells (Fig. 5c, d) to 30 µg/mL AGE-modified bovine serum albumin (glucose-derived, glycolaldehyde-mediated AGE-BSA) for 15 min caused rapid association between MD2 and TLR4. Compared with AGE-BSA, however, neither MGO-BSA, nor CML-BSA at 30 µg/mL induced MD2-TLR4-MyD88 interaction in MPMs and H9C2 cells (Supplementary Fig. 23a–c).

Subsequently, incubation with AGE-BSA, but not MGO-BSA and CML-BSA for 1 h induced I-κB degradation in MPMs and H9C2 cells (Supplementary Fig. 23c). AGE-BSA also induced TNF-α and IL-6 in MPMs harvested from WT mice (Fig. 5e) consistent with activated MD2-TLR4. The level of TNF-α and IL-6 in MPMs isolated from MD2KO mice was significantly less compared with cells from WT mice, confirming MD2 involvement. Similarly, direct exposure of H9C2 cells to AGE-BSA induced a threefold increase in

*Tnfa* and *Il6* (Fig. 5f), which was blocked in cells transfected with MD2 siRNA. As expected, MGO-BSA and CML-BSA at 30 µg/mL failed to induce *Tnfa* and *Il6* in MPMs (Supplementary Fig. 23d). These results indicate that HG-derived AGE products activate MD2 and induce inflammation.

**AGEs directly bind MD2 and activate MD2-TLR4.** The above data suggest that HG-generated AGE products directly bind to and activate MD2-TLR4 signaling complex. We investigated this idea using a combination of approaches. We first exposed H9C2 cells to HG for 5-15 min and show rapid pull-down of AGE using either MD2 or TLR4 as the precipitating antibody (Fig. 6a). H9C2 cells challenged directly with purified AGE-BSA also showed the formation of AGE-MD2-TLR4 complex in a rapid fashion (Fig. 6b). Interestingly, MD2 knockdown reduced the level of AGEs associating with TLR4 (Fig. 6c), implicating MD2 as a required binding partner for AGEs. In support of this finding, TLR4 knockdown showed no effect on the HG-induced AGE-MD2 interaction (Fig. 6d). Pretreatment of cells with MD2 inhibitor L6H21 also prevented pull-down of AGEs following MD2 or TLR4 antibody precipitation in HG-challenged H9C2 cells (Fig. 6e). Although this co-IP data supports direct interactions of AGE products with MD2, it cannot rule out participation by other molecules.

We then investigated AGE-MD2 interactions under cell-free conditions using ITC and ELISA. ITC measurements of rhMD2 reacting with AGE-BSA indicated direct interaction, producing a Kd value of 81 µM (Fig. 6f). A sandwich ELISA based on AGE- and MD2-antibodies was used to confirm ITC data. Results indicated that addition of purified AGE-BSA and rhMD2 (1:1 or 1:0.5) in PBS correlated with detection of AGE-MD2 complex (Fig. 6g). Thus, the data obtained cell-based and cell-free studies provide strong evidence that AGEs directly bind MD2, which would form a heterotrimer of AGE-MD2-TLR4 on the cell surface, leading to subsequent activation of the pro-inflammatory signaling cascades.

Although the AGE-BSA preparation used in our study contained very low endotoxin levels, we explored the possibility that MD2-TLR4 activation in response to AGE-BSA may have resulted from endotoxin contamination. To test this, we exposed H9C2 cells to AGE-BSA with or without polymyxin B. Polymyxin B binds to lipid A of LPS and will mitigate any false-positive results from endotoxin contamination. Our results show that presence of polymyxin B does not alter AGE-BSA-induced MD2-TLR4 complex formation or induction of *Tnfa* and *Il6* in H9C2 cells (Supplementary Fig. 24).

We also tested the possibility that the prototypical receptor for AGE (RAGE) may contribute to HG- and AGE-induced inflammatory responses in cardiac cells. We knocked down the expression of RAGE in H9C2 cells using shRNA and achieved 90% silencing (Supplementary Fig. 25a). We found that silencing RAGE does not affect HG-induced MD2-TLR4 complex formation (Supplementary Fig. 25b). However, RAGE silencing reduced the level of cytokine induction by HG, suggesting that RAGE activation may also contribute to the inflammatory responses (Supplementary Fig. 25c, d). Similarly, challenging cells with AGE-BSA did not alter AGE-MD2-TLR4 interaction upon RAGE knockdown (Supplementary Fig. 25e) but prevented the full induction of cytokines (Supplementary Fig. 25f, g). These results indicate that under our experimental conditions, HG and AGE products activate distinct MD2-TLR4 and RAGE pro-inflammatory signaling pathways.

**Elevated AGE products and MD2-AGE complexes in diabetes.** Our findings from cellular and molecular studies suggested that one mechanism by which HG causes cardiac cell injury is by direct AGE interaction with MD2 to activate MD2-TLR4 signaling complex. To explore this complex formation in animal models of diabetes and in human subjects, we first determined the levels of AGE products. Measurement of AGE content in myocardial tissues of STZ-induced type 1 diabetic mice showed a twofold increase over levels in non-diabetic controls (Fig. 7a). Serum levels of AGE products were also increased in WT-STZ mice (Supplementary Fig. 26a). No changes in AGE content was observed in MD2KO-STZ mice compared with WT-STZ mice. Consistent with our cell culture studies, no increases in MGO and CML levels were noted in control and diabetic mice (Supplementary Fig. 26b-c). MD2 antibody pulled down increased amounts of AGE and TLR4 in lysates prepared from WT-STZ mice (Fig. 7b), suggesting increased MD2-AGE and MD2-TLR4 complex formation. Since MD2 also exists in a soluble form (sMD2) and appears to be important for sensing endogenous ligands[31,32], we speculated that circulating sMD2 and AGEs may form complexes in the context of diabetes. We detected serum sMD2-AGEs complexes using a sandwich ELISA and show approximately twofold higher content of MD2-AGE complexes in WT-STZ mice compared with WT-Con (Fig. 7c). Parallel analyses were made in a type 2 diabetic mouse model (*db/db*). Analysis of cardiac tissues in *db/db* mice showed increased AGE levels (Fig. 7d), enhanced MD2-AGE and MD2-TLR4 complexes (Fig. 7e), and MD2-AGE complexes in serum (Fig. 7f), compared with non-diabetic *db/m* controls. Furthermore, as with the STZ-induced model of type 1 diabetes, myocardial tissue of *db/db* mice showed increased levels of TNF-α, IL-6, and MD2 levels compared with *db/m* controls (Supplementary Fig. 27a–c).

We wanted to explore whether enhanced MD2-AGE interaction is evident in human subjects with diabetes. Blood was collected from healthy subjects, and diabetic subjects with or without evidence of DCM. We isolated peripheral blood mononuclear cells (PBMCs) and prepared serum. Analysis of serum content of AGE products showed 2-fold greater levels in DCM subjects compared with healthy subjects (Fig. 7g). Interestingly, we observed that HG addition to healthy human serum significantly induces AGE production within 5 min (Supplementary Fig. 28a), which is consistent with our finding that HG rapidly produces AGE in culture media containing serum (Fig. 5a–d). In addition to increased serum AGE levels, PBMCs isolated from diabetic subjects with DCM or without cardiomyopathy showed elevated levels of MD2 proteins (Supplementary Fig. 28b), and increased AGE-MD2-TLR4 complexes (Fig. 7h). Analysis of serum samples further showed elevated soluble MD2 (Supplementary Fig. 28c), inflammatory cytokines TNF-α and IL-6 (Supplementary Fig. 28d–e), and MD2-AGE complexes (Fig. 7i) in diabetic subjects with DCM and without cardiomyopathy, compared with healthy samples. Our identification of increased MD2-AGE complexes in diabetic mice and human subjects extended the findings obtained from in vitro studies, and strengthen the idea that direct AGE-MD2 interactions are the initiating events driving the cardiac inflammatory state in diabetes.

## Discussion

Diabetic patients commonly succumb to cardiovascular complications. Yet, the mechanisms by which high circulating glucose levels cause diabetic inflammatory tissue injury remain poorly understood. Experimental evidence shows an overactive TLR4 signaling pathway, linking MyD88 with the activation of MAPK and NF-κB and upregulation of proinflammation factors, as a critical factor in diabetic myocardial injury and dysfunction[9,19–22,33]. Our study focused on understanding how

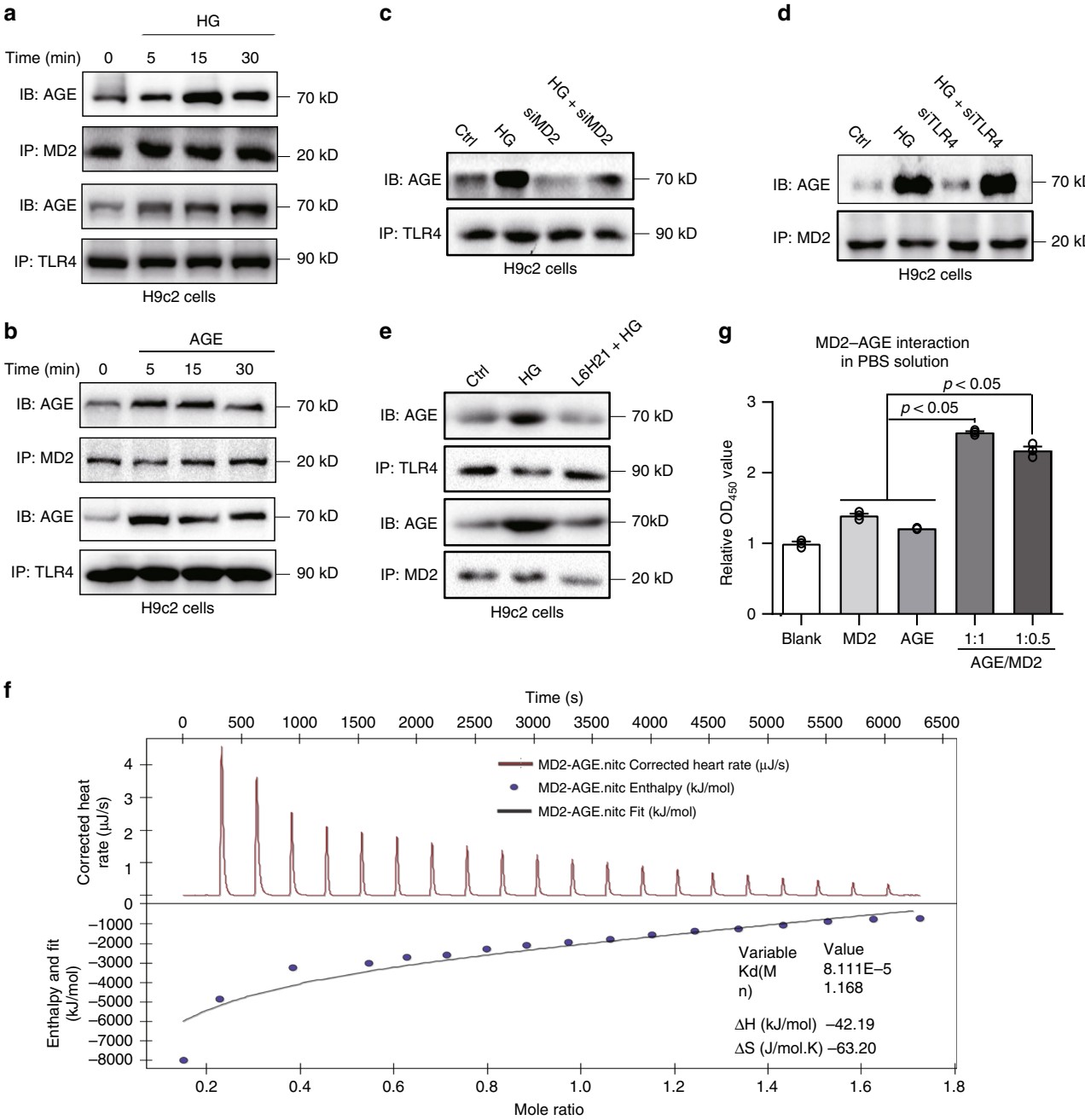

**Fig. 6 AGE products bind to MD2. a** Representative immunoblot showing co-immunoprecipitation of AGE-MD2 and AGE-TLR4 complexes in H9C2 cells exposed to HG (33 mM glucose) for indicated times [$n = 3$]. **b** Representative blots of co-immunoprecipitated AGE-MD2 and AGE-TLR4 complexes in H9C2 cells challenged with 33 μg/mL AGE-BSA for indicated times [$n = 3$]. **c** Representative blots of co-immunoprecipitated AGE-TLR4 complexes in H9C2 cells transfected with MD2 siRNA (siMD2) and exposed to HG (33 mM glucose) for 5 min [$n = 3$]. **d** Representative blots of co-immunoprecipitated AGE-TLR4 complexes in H9C2 cells transfected with TLR4 siRNA (siTLR4) and exposed to HG (33 mM glucose) for 5 min [$n = 3$]. **e** Co-immunoprecipitation of AGE-MD2 and AGE-TLR4 complexes in H9C2 cells pretreated with 10 μM L6H21 for 30 min and then challenged with HG (33 mM glucose) for 5 min [$n = 3$]. **f** Isothermal titration calorimetry analysis of interactions between AGE-BSA and rhMD2. Representative image was shown from three independent experiments. **g** Sandwich ELISA analysis of AGE-MD2 interaction. AGE-BSA and rhMD2 proteins were added at ratios of 1:1 or 1:0.5, or each alone to bovine AGE ELISA plates. Complexes were detected by anti-human MD2 antibody and TMB chromagen [means ± SEM; $n = 3$ examinations]. Source data are provided as a Source Data file. P-values by one-way ANOVA in **g** followed by Tukey's post hoc test are indicated.

glucose activates the TLR4 signaling cascade and induces inflammatory responses. Our findings support a working model in which high levels of glucose induce an AGE-dependent activation of the MD2-TLR4 immune signaling complex in macrophages and cardiomyocytes, resulting in inflammatory factor expression and myocardial injury (Fig. 8).

Increased levels of TLR4 and downstream accessory proteins have been reported to be increased in human diabetes and in experimental models. Specifically, studies have shown the HG induces TLR4 in myocytes[11] and monocytes[12]. Diabetic mice and rats have also been reported to express upregulated TLR4 in heart tissues[17,18]. Studies have also shown that oxidative stress may

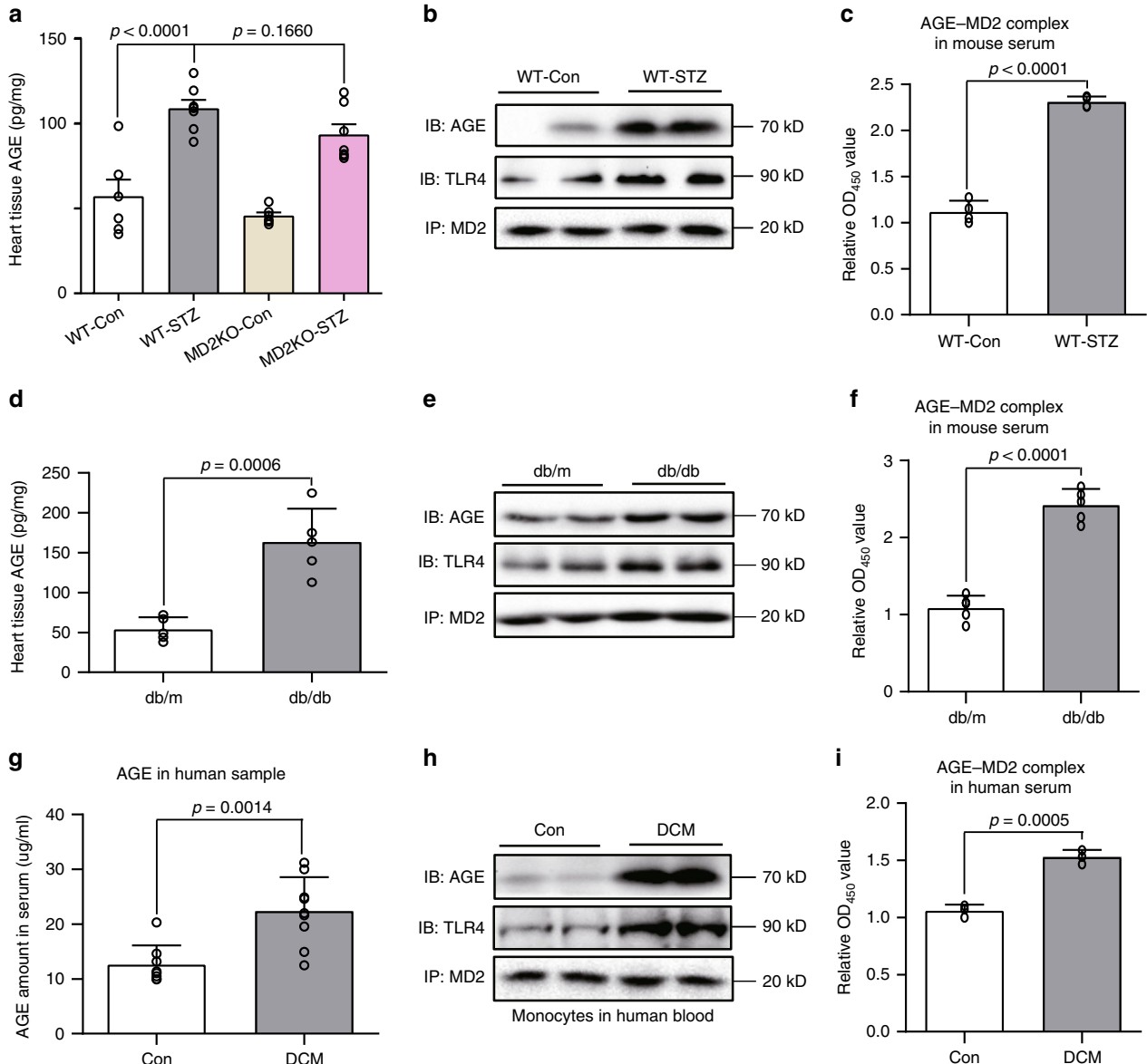

**Fig. 7 AGE-MD2 complexes in serum and cardiac tissues in diabetes. a** Levels of AGE products in heart tissues of type 1 mouse model of diabetes. C57BL/6 wild-type and MD2KO mice were made diabetic by streptozotocin. Heart tissues were harvested at 16 weeks and levels of AGE products were determined by ELISA [experimental groups are as described in Fig. 2; means ± SEM; $n = 6$ per group]. **b** Representative blots showing co-immunoprecipitation of MD2-AGE complexes in heart tissues from type 1 mouse model of diabetes. Tissues from WT-Con and WT-STZ mice at 16 weeks after confirmation of diabetes were examined [$n = 6$; two samples per group shown]. **c** MD2-AGE complexes were measured in serum of WT-Con and WT-STZ mice at 16 weeks [means ± SEM; $n = 4$]. **d** Levels of AGE products in heart tissues of type 2 mouse model of diabetes. Heart tissues from *db/m* (controls) and *db/db* (diabetic) mice were harvested at 16 weeks. AGE products were determined by ELISA [means ± SEM; $n = 5$ per group]. **e** Representative blots showing co-immunoprecipitation of MD2-AGE complexes in heart tissues from type 2 mouse model of diabetes [experimental groups are as shown in panel D; $n = 6$; two samples per group shown]. **f** MD2-AGE complexes were measured in serum of *db/m* and *db/db* mice at 16 weeks [means ± SEM; $n = 5$]. **g** Serum levels of AGE products in healthy human subjects and diabetic subjects with cardiomyopathy [Co = healthy subjects ($n = 8$), DCM = diabetic subjects with cardiomyopathy ($n = 9$); means ± SEM]. **h** Representative blots showing AGE-MD2 complexes in human blood mononuclear cells isolated from healthy subjects (Con) and diabetic subjects [$n = 6$; two samples per group shown]. **i** MD2-AGE complexes in serum samples from human subjects [means ± SEM; $n = 3$ per group]. Source data are provided as a Source Data file. *P*-values by one-way ANOVA in a followed by Tukey's post hoc test are indicated. *P*-values by unpaired *t* test are indicated in **c**, **d**, **f**, **g** and **i**.

increase TLR4 proteins[12]. What has remained elusive, however, is how TLR4 is activated by glucose or other factors in diabetes. Our studies have demonstrated a rapid mechanism of direct activation of TLR4 by HG. Importantly, our studies have identified MD2 as a requisite for TLR4 activation. In diabetic MD2 knockout mice or diabetic mice administered L6H21, NF-κB, and MAPK pathways were inhibited, and production of pro-inflammatory molecules was suppressed, indicating that MD2 deficiency

correlated with suppressed inflammation. Moreover, diabetes-induced myocardial fibrosis, cardiomyocyte hypertrophy, and cardiac dysfunctional was effectively prevented with MD2 deficit. These results strongly suggest that MD2 plays a critical role in initiating diabetes-associated inflammatory injury leading to functional and structural deficits in the heart.

Our studies show that TLR4 is activated when HG-derived AGE products directly engage the TLR4 pathway (Fig. 8). AGE

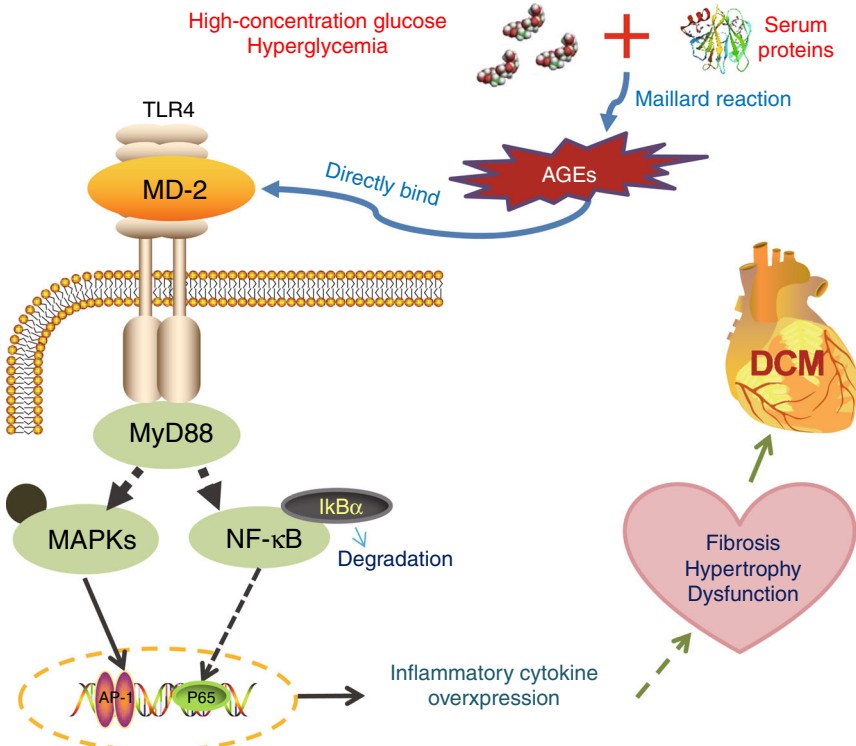

**Fig. 8 Working model of AGE-induced MD2-TLR4 activation in diabetes.** Schematic illustration showing the key findings of the study. High levels of glucose generate AGE products in the extracellular environment. AGE products bind directly to MD2 and lead to activation of the immune signaling complex MD2-TLR4. Intracellular adaptor proteins such as myeloid differentiation primary response protein-88 (MyD88) are recruited to AGE-MD2-TLR4 complex. TLR4 then leads to activation of mitogen-activated protein kinase (MAPK) and NF-κB signaling pathways, and regulation of genes involved in inflammatory and tissue remodeling responses.

products are generated by non-enzymatic glycation/oxidation of proteins, lipids or nucleotides which accumulate in the extracellular matrix, both in tissues as well as in the circulation[34]. AGE products are known to signal predominantly through the prototypical RAGE, but can also serve as ligands for other receptors including macrophage scavenger receptor type II and CD36[35]. Our studies support an alternate signaling pathway to RAGE, in which AGE products bind and activate the MD2-TLR4 immune signaling complex. In addition, cell-free studies using purified MD2 and AGE-BSA corroborated direct AGE binding to MD2. Moreover, both MD2-TLR4 and RAGE appeared to contribute to HG- and AGE-induced pro-inflammatory factor induction. It is unknown whether RAGE and MD2-TLR4 are completely distinct pathways or play an interacting role. Recent studies utilizing high mobility group box 1 (HMGB1) which is a ligand for both TLR4[36] and RAGE[37], offers some insight. HMGB1 and AGE products can activate MyD88 through RAGE[38]. As MyD88 appears to be shared between TLR4 and RAGE, it is possible that MD2-TLR4 and RAGE crosstalk to fully engage MyD88 in cardiomyocytes. However, since we are able to completely normalize inflammatory cytokine production in response to HG and AGE by blocking MD2 (L6H21, MD2 siRNA, or in MD2KO cells), we believe that MD2-TLR4 pathway plays a pivotal role in inflammatory responses to HG and AGE. This would suggest that even RAGE activity would require TLR4 activation. Indeed, this notion is supported by recent studies by Valencia and colleagues who reported that AGE products fail to bind RAGE and upregulate VCAM-1 or TNF-α in endothelial and mononuclear cells unless TLR4 was activated[39]. Future studies utilizing cardiomyocytes and macrophages with RAGE knockout and selective engagement of TLR4 versus RAGE would be needed to confirm these ideas.

Another important finding in our study was increased levels of AGE-MD2 complexes in diabetic human and mouse serum. Diabetic conditions are causally linked to elevated circulating AGEs[40], and this was further corroborated by our study in which elevated AGE products were detected in serum of human subjects with DCM, and mouse models of type 1 and 2 diabetes. Moreover, MD2 is a secreted protein and its soluble form, sMD2, is found in the circulation[32,41]. Elevated circulating sMD2 levels are associated with patients with severe infections, such as sepsis and rheumatoid arthritis[41]. Although the exact function of sMD2 is not fully defined, studies have shown that complex formation between sMD2 and LPS increases circulating MD2 protein half-life[42] and that this complex interacts with TLR4-expressing cells to confer LPS sensitivity[32]. Interestingly, even without interacting with LPS, sMD2 is able to bind cell-surface TLR4[43]. Therefore, the consensus is that sMD2 facilitates innate immune responses, in part through activating TLR4 in MD2-deficient cells. Although sMD2 level was only measured in human serum samples in our study, we did detect significantly elevated AGE-MD2 complexes in serum of diabetic subjects, and in both models of diabetic mice. In addition, myocardial tissues from diabetic mice and PBMCs from human subjects showed elevated AGE-MD2 complexes. At this juncture, we do not know if AGE-MD2 complexes were formed during transit in the blood or released by tissues or circulating cells. Nonetheless, such elevated levels of pre-formed AGE-MD2 in circulation and tissues under diabetic conditions suggest that this could be one reason why TLR4 was predominantly activated by AGE.

One of the biggest unanswered question in our study is the mode by which AGE products bind to MD2. We attempted to explore the potential binding mode using a combination of cell-based and cell-free assays. Until now, only the structure of LPS-

MD2 complex has been reported by crystallography[44]. To gain some insight into how AGE products may interact with MD2, we conducted competition studies using LPS in H9C2 cells. Using a functionally inactive LPS (*Rhodobacter sphaeroides* LPS, RS-LPS) which binds to MD2[45], we found that HG-induced MD2-TLR4 complex formation was suppressed by ~50% by RS-LPS (Supplementary Fig. 29a). Furthermore, HG partially displaced H9C2 cell surface-bound FITC-labeled LPS (Supplementary Fig. 29b). The ability of HG to partially compete with LPS binding suggests that AGE products may share some common binding sites on MD2 with LPS. However, this notion needs to accompany a cautionary statement. Considering that AGE products are highly heterogenous in composition, the exact binding site(s) or binding modes of AGE products with MD2 will be highly challenging to delineate. We know that different AGE products may be generated depending on the glucose concentration used, the reacting process and glycation duration[46]. Experimental studies also show that glucose- and glycolaldehyde-modified BSA is significantly different compared with glyoxal- and methylglyoxal-modified BSA, in exhibiting fluorescence patterns, CML content, and uptake patterns in macrophages[30]. Considering all of these variables, it is plausible that MD2 acts as a scavenger co-receptor and interacts with various modified/altered proteins with different affinities. In support of this idea is our recent discovery that MD2 binds to saturated fatty acids and oxidized lipoproteins to initiate TLR4 signaling[23,25]. Future studies aimed at determining whether ligands with different affinities induce different intracellular signaling arms of TLR4 and when signaling thresholds are met will greatly enhance our understanding.

It is important to point out a few limitations of our study. First, we are unable to conclusively rule out the involvement of MGO- and CML-modified proteins. We show that MGO and CML levels in vitro and in vivo, detected using published methods, are not increased at the time points examined. In addition, stimulation with MGO-BSA and CML-BSA could not activate MD2/TLR4 pro-inflammatory response in cells. Further studies are certainly needed to fully understand the in vivo pathways of AGE formation in the extracellular and intracellular milieu, as well as identification of the proteins, lipids or nucleotides glycated, since glycated molecules are physiologically impaired[34,47]. Similarly, it is important to understand that clinical diabetic cardiomyopathy takes years to develop. Experimentally, however, acute or short-term exposure to HG is enough to generate the readout, which may not unmask all AGE moieties involved. Second, due to the limited number of samples from diabetic subjects, we were unable to complete the full spectrum of analyses that were conducted in experimental mice and cultured cells. Although key readouts were confirmed in serum and cell samples from diabetic subjects, further studies in samples from human diabetic subjects would be important.

In summary, the following key findings support an AGE-activated MD2-TLR4 pathogenic mechanism as a central driving force behind inflammatory myocardial injury in diabetes: (1) HG was associated with elevated AGE and AGE-MD2 complexes in serum and myocardial tissue; (2) diabetic human subjects and mice showed upregulated MD2 and TLR4 levels; (3) AGE products bound directly to MD2 and activated MD2-TLR4 signaling; (4) MD2 knockout or pharmacological inhibition effectively blocked diabetes-induced NF-κB/MAPK activation along with upregulation of inflammatory factors. These findings provide mechanistic insight in which AGE-activated MD2-TLR4 orchestrates adverse cardiac tissue remodeling in diabetes, and underscore MD2 as a potential target for DCM.

## Methods

**Reagents**. D-glucose, mannitol, lipopolysaccharide (LPS, from *Salmonella typhosa*), fluorescein isothiocyanate-conjugated LPS (FITC-LPS, from *E. coli* 055:B5),

*Rhodobacter sphaeroides* LPS (RS-LPS), DAPI, 2-DG, phalloidin, Polymyxin B, and streptozotocin (STZ) were purchased from Sigma (St. Louis, MO). Recombinant human MD2 (rhMD2) protein was purchased from R&D Systems (Minneapolis, MN, USA). Glucose-derived AGE-modified BSA (AGE-BSA) with a purity >98% was purchased from BioVision Incorporated (Cat # 2223; Milpitas, CA). Lots used in the study had a reported endotoxin level <0.1 EU/µg. Endotoxin contamination was verified through Chromogenic TAL Endpoint Assay Kit (Cat # EC64405, Bioendo Technology Co., LTD, Xiamen, China), before using AGE-BSA in any experiments. CML-BSA was purchased from Cayman Chemicals (Cat # 22972; Ann Arbor, MI). MGO-BSA was purchased from StressMarq Biosciences Inc (Cat #SPR-208B; Victoria, BC, Canada). Glycolytic function in cells was measured by Agilent Seahorse XF Glycolysis Stress Test Kit (Agilent Technologies Co. Ltd, Beijing, China).

Anti-GAPDH (Clone: G-9, cat no. sc-365062), anti-IκBα (Clone: H-4, cat no. sc-1643), anti-MyD88 (Clone: E-11, cat no. sc-74532), anti-ERK1/2 (Clone: C-9, cat no. sc-514302), anti-p-ERK1/2 (Clone: 12D4, cat no. sc-81492), anti-F4/80 (Clone: C-7, cat no. sc-377009), anti-TGF-β (Clone: 3C11, cat no. sc-130348), anti-Col 1 (Clone: sc-59772), anti-TLR4 (Clone: 25, cat no. sc-293072), anti-MD2 (Clone: J-12B, cat no. sc-80183), anti-GLUT4 (Clone: IF8, cat no. sc-53566), anti-Flag tags (Clone: G-8, cat no. sc-166384), anti-HA tags (Clone: F-7, cat no. sc-7392), anti-ANP (Clone: F-2, cat no. sc-515701), anti-MyHC (Clone: B-5, cat no. sc-376157), anti-MMP-2 (Clone: 8B4, cat no. sc-13595), anti-MMP-9 (Clone: 6-6B, cat no. sc-12759), anti-ICAM-1 (Clone: G-5, cat no. sc-8439), anti-VCAM-1 (Clone: E-10, cat no. sc-13160), anti-α-actin (Clone: 1A4, cat no. sc-32251), anti-CD68 (Clone: E-11, sc-17832), anti-TNF-α (Clone: 52B83, cat no. sc-52746), and anti-RAGE (Clone: E-1, cat no. sc-74473) antibodies were purchased from Santa Cruz Biotechnology (Santa Cruz, CA). Antibodies directed against JNK (cat no. 9252) and p-JNK (Clone: G9, cat no. 9255), P38 (cat no. 9212) and p-P38 (Clone: D13E1, cat no. 8690) were obtained from Cell Signaling (Danvers, MA). Anti-AGE antibody (cat no. ab23722) was obtained from AbCam. Compound L6H21, a selective MD2 inhibitor, was synthesized, validated, and prepared with a purity of 98.9% as described by us[27]. Our previous studies have established anti-inflammatory activities of L6H21[27]. L6H21 was dissolved in DMSO for in vitro studies and in 1% sodium carboxyl methyl cellulose (CMC-Na) for in vivo studies. MGO levels in culture media and tissues were determined as described previously[48]. CML levels were measured by CircuLex CML/Nε-Carboxymethyl-lysine ELISA (cat no. CY-8066; Clinisciences, Nanterre, France).

**Cell isolation culture**. An embryonic rat heart-derived H9C2 line was obtained from the Shanghai Institute of Biochemistry and Cell Biology (Shanghai, China) and cultured in DMEM (Gibco, Eggenstein, Germany) containing 1 g/L glucose supplemented with 10% heat-inactivated FBS (Thermo Fisher, Waltham, MA), 100 U/ml of penicillin, and 100 mg/ml of streptomycin. Mouse primary peritoneal macrophages (MPMs) were isolated from C57BL/6 wild-type mice or MD2[-/-] mice as described by us previously[23]. Primary rat cardiomyocytes were isolated from Sprague-Dawley (SD) rats as described previously[23]. Cardiac endothelial and fibroblasts were isolated from C57BL/6 wild-type mice used the methods described previously[49,50]. All cells were cultured and expanded in media containing heat-inactivated serum. Where indicated, serum which did not undergo heat-inactivation (Thermo Fisher) was used. To determine the effects of high levels of glucose, cardiomyocytes, H9C2 cells, and MPMs were exposed to 33 mM glucose as shown by us[51,52] and others[53,54], with the exception of dose–response studies.

**Human subject recruitment**. Eight apparently healthy subjects without diabetes, and seven diabetic subjects without evidence of suppressed ejection fraction (EF%), and nine diabetic patients with reduced EF% and without the indication of other cardiac diseases were recruited from the hospital clinic and community, based on an a priori protocol. All subjects were screened for the echocardiographic exam. DM and DCM candidates were excluded if they were pregnant, had a self-reported history of symptomatic micro- or macrovascular complications of diabetes (including nephropathy, neuropathy, retinopathy, peripheral vascular disease, ischemic heart disease, and stroke) or other significant comorbidities including malignancy, renal failure, or significant psychiatric illness. The information of healthy subjects and diabetic patients are shown in the Supplementary Table 1. Consent was obtained from patients to disclose age and sex, and location of treatment. Approval for this study was granted by the Human Research Ethics Committees of Wenzhou Medical University, Wenzhou, China. All subjects gave written, informed consent to participate in the study and research was carried out in accordance with the Declaration of Helsinki (2008) of the World Medical Association. Serum was prepared by centrifugation of fresh blood samples at $1000 \times g$ for 20 min. About 5 mL of blood was obtained from each donor using BD Vacutainer tubes containing acid-citrate-dextrose anti-coagulant, solution A (ACD-A; BD Sciences), from which plasma and peripheral blood mononuclear cell (PBMCs) were isolated.

**Animal care and preparation**. Male C57BL/6 mice (18–22 g) and male SD rats (200–220 g) were obtained from the Animal Centre of Wenzhou Medical University (Wenzhou, China). MD2[-/-] mice on C57BL/6 background were purchased from Riken BioResource Center (Tokyo, Japan). Male *db/db* (C57BLKS/J-leprdb/leprdb) mice and male *db/m* littermates were purchased from Model Animal Research Center of Nanjing University (Nanjing, China). All animals were housed at a constant room

temperature with a 12:12 h light–dark cycle and fed with a standard rodent diet and water in the Animal Centre of Wenzhou Medical University. All animals received humane care according to the National Institutes of Health (USA) guidelines. All animal care and experimental procedures were approved by the Wenzhou Medical University Animal Policy and Welfare Committee.

To model cardiac injury in type 1 diabetes, C57BL/6 (WT) and MD2$^{-/-}$ (MD2KO) mice were made diabetic by a single dose intraperitoneal injection of streptozotocin (STZ, 100 mg/kg in citrate buffer, pH 4.5). This dose of STZ has been shown to reliably induce diabetes in C57BL/6 mice[55], and validated by our group[51,52]. Vehicle control group of mice received the same volume of citrate buffer. Fasting blood glucose levels were measured starting day 3 using glucometer. Mice showing fasting glucose levels >12 mM (~216 mg/dL) on 3 consecutive days were considered diabetic, as outlined previously[56]. Body weights and fasting glucose levels were measured in all mice weekly for the duration of the study. The experimental groups were as follows: non-diabetic WT controls (WT-Con, $n = 6$), STZ-induced WT diabetic mice (WT-STZ, $n = 7$), non-diabetic MD2$^{-/-}$ controls (MD2KO-Con, $n = 6$), and STZ-induced MD2$^{-/-}$ diabetic mice (MD2KO-STZ, $n = 7$). All other mice were euthanized 16 weeks following the onset of diabetes. Euthanasia was performed under sodium pentobarbital anesthesia and the blood was collected for subsequent analyses. Heart tissues were collected and fixed in 4% paraformaldehyde for pathological analysis and/or snap-frozen in liquid nitrogen for gene and protein expression analyses. Systolic and diastolic cardiac function was determined non-invasively by transthoracic echocardiography in anesthetized mice, 1 day before the sacrifice. Mice were anesthetized using isoflurane and echocardiography was performed by SONOS 5500 ultrasound (Philips Electronics, Amsterdam, Netherlands) with a 15-MHz linear array ultrasound transducer.

To examine cardiac MD2-TLR4 activation in a type 2 model of diabetes, we utilized 8-week-old male *db/db* mice. *db/m* littermates were used as controls. Male *db/db* mice ($n = 5$) and *db/m* mice ($n = 5$) were fed the standard rodent diet for 16 weeks. Mice were then euthanized under sodium pentobarbital anesthesia and the blood and heart tissue were collected for subsequent analyses.

**Bone marrow transplantation**. One week before transplantation, MD2$^{-/-}$ mice (MD2KO) were put in filter-top cages and given acidified water containing neomycin (1.1 mg/L) and polymyxin B sulfate (1000 U/L). One day prior to transplantation, mice were subjected to total body irradiation (6 Gy). For transplantation, MD2KO mice were injected intravenously with $5 \times 10^6$ bone marrow cells from pools of bone marrow from C57BL/6 (WT) and MD2$^{-/-}$ (MD2KO) mice. Tail-clip samples and peritoneal macrophages were used for genotyping and confirmation of reconstitution.

**Histology and tissue staining**. Heart tissues were fixed in 4% paraformaldehyde, embedded in paraffin and sectioned at 5-μm thickness. After dehydration, sections were stained with hematoxylin and eosin (H&E). Stained sections were evaluated for general histopathological damage using light microscopy (Nikon, Japan). Cardiomyocyte size was evaluated in cross-sections of heart tissues. Additional paraffin sections were stained with Masson's Trichrome (Beyotime Biotech, Nantong, China) and Sirius Red (Beyotime Biotech, Nantong, China) to assess cardiac fibrosis. The stained sections were viewed under a light microscope (Nikon, Japan). Paraffin sections were also used to perform immunohistochemistry for TNF-α and CD68 using routine techniques. Immunoreactivity was detected by diaminobenzidine (DAB).

For fluorescence staining, harvested mouse hearts were embedded in OCT compound, and cut into 5-μm-thick sections. Sections were incubated with primary mouse antibodies directed against MD2, and α-actin or F4/80 in a humidified chamber at 4 °C overnight. The sections were subsequently incubated with anti-mouse and anti-rabbit secondary antibodies for 1 h in the dark.

**Enzyme-link immunosorbent methods**. Pro-inflammatory cytokines TNF-α and IL-6, and AGE products in cell culture medium, homogenized mouse heart tissue, or serum were determined using commercially available ELISA kits. ELISA for TNFα and IL-6 were from eBioscience (San Diego, CA). ELISA for human AGE (ml-024019) and mouse AGE (ml-002154) were from mlbio (Shanghai Enzyme-linked Biotech, Shanghai, China). Total amount of TNFα, IL-6, or AGE in the cell culture medium was normalized to the total amount of protein in the viable cell pellets. Experiments were performed in triplicates.

To detect AGE-MD2 complexes in serum samples, a sandwich ELISA was used. Anti-human or anti-mouse AGE antibodies (Shanghai Enzyme-linked Biotech, Shanghai, China) were pre-coated onto 96-well plates. Serum samples were added and incubated for 1 h at 37 °C. Plates were washed six times and incubated with anti-human MD2 antibody (Abcam) or anti-mouse MD2 antibody (Abcam) for 1 h at 37 °C. Antigen-antibody complexes were washed six times and incubated with peroxidase-conjugated secondary antibody (Yeasen Bio) for 1 h at 37 °C. TMB substrate was added, followed by a stop solution (mlbio), and the horseradish peroxidase activity was measured in M5 microplate reader at 450 nm. Values are reported in optical density units.

To determine binding between AGE-BSA and recombinant human (rh) MD2, individual protein solutions or mixed solutions at 1:1 or 1:0.05 ratios were loaded

on plates coated with bovine AGE antibody. Plates from commercially available bovine AGE ELISA (ml-003305; mlbio) were used. Plates were washed and incubated with anti-human MD2 antibody (Abcam) for 1 h at 37 °C. Detection was performed as described above.

**Gene silencing**. MD2, TLR4, and GLUT4 were silenced in H9C2 cells through siRNA. Rat MD2 siRNA (5′-TATTAAATACTGTTGC-3′), rat TLR4 siRNA (5′-TTCTAACTTCCCTCCT-3′), and rat GLUT4 siRNA (5′-CAGAGCUA-CAAUGCAACUUTT-3′) were purchased from Gene Pharma Co. Ltd. (Shanghai, China). Negative control transfections included scrambled siRNA sequences. RAGE in H9C2 cells was silenced by transfecting cells with plasmids encoding RAGE shRNA (Sailan Biotech, Hangzhou, China). All transfections were carried out using LipofectAMINE™ 2000 (Invitrogen, Carlsbad, California). Knockdown of genes in the transfected cells was confirmed by western blot analysis.

**Western blot and immunoprecipitation**. Collected cells or homogenized heart tissues were lysed and protein concentration determined by the Bradford assay. Lysates were separated by sodium dodecyl sulfate-polyacrylamide gel electrophoresis, and electro-transferred to a nitrocellulose membrane. The membranes were blocked for 1.5 h at room temperature in Tris-buffered saline (pH 7.6) containing 0.05% Tween 20 and 5% non-fat dry milk. Primary antibody incubations were carried out at 4 °C overnight. Secondary antibodies were applied for 1 h at room temperature. Immunoreactivity was visualized using enhanced chemiluminescence reagents (Bio-Rad Laboratories, Hercules, CA), and quantified using Image J analysis software version 1.38e (NIH, Bethesda, MD, USA). Values were normalized to respective protein controls.

Protein complexes were evaluated by co-immunoprecipitation combined with immunoblotting. Cell lysates (300–500 μg) were incubated with precipitating antibody at 4 °C overnight, and immunoprecipitated with protein A + G-sepharose beads with shaking at room temperature for 2 h. The protein–bead complexes were washed five times with PBS, electrophoresed, transferred to PVDF membranes and detected with immunoblotting antibody.

**Real-time quantitative PCR**. Cells and heart tissues were homogenized in TRIZOL (Invitrogen, Carlsbad, CA). Both reverse transcription and quantitative PCR (qPCR) were carried out using a two-step M-MLV Platinum SYBR Green qPCR SuperMix-UDG kit (Invitrogen, Carlsbad, CA). Eppendorf Mastercycler® Ep Realplex detection system (Eppendorf, Hamburg, Germany) was used for qPCR analysis. The primers of target genes are listed in the Supplementary Table 2 and were obtained from Invitrogen (Shanghai, China). The amount of each gene was determined and normalized to the amount of β-actin.

**Isothermal titration calorimetry (ITC) assay**. The interactions between MD2 and AGE products or glucose were carried out on a Nano-ITC instrument (TA instruments, New Castle, DE) at 25 °C. The proteins were dissolved in ddH$_2$O. AGE-BSA (1 μM) or rhMD2 (10 μM) were added to the calorimetric reaction cell, and an injection syringe was filled with MD2 protein solution (5 μM) or D-glucose (100 μM), respectively, with stirring at $18 \times g$. Each titration experiment was performed with 20 injections of 2.5 μL at 300 s equilibration intervals. The heat of dilution for AGE-BSA or MD2 protein was determined by titrating it into the ddH$_2$O. Data were fitted with the NanoAnalyze software package (TA Instruments). The total heat exchanged during each injection of MD2 to AGE-BSA or glucose to MD2 was fitted to an independent model with variable parameters.

**Statistical analysis**. All experiments were randomized and blinded. In vitro experiments were repeated at least three times. Data were presented as means ± SEM, and individual data points are plotted in figures. The statistical significance of differences between groups was obtained by the unpaired student's *t*-test or one-way ANOVA followed by Tukey's post hoc test in GraphPad Pro5.0 (GraphPad, San Diego, CA). Details of statistical testing can be found in the figure legends and in the source data file. Differences were considered to be significant at $p < 0.05$.

**Reporting summary**. Further information on experimental design is available in the Nature Research Reporting Summary linked to this paper.

## Data availability
The source data underlying Figs. 1–7 in the article and Supplementary Figs. 1–29 in the Supplementary Information are provided as a Source Data file. All other data are included within the article or Supplementary Information or are available from the corresponding author upon reasonable request.

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

## Acknowledgements

Professor Guang Liang takes full responsibility of the work, including the study design, access to data, and the decision to submit and publish the manuscript. Financial support was provided by the National Key Research Project of China (2017YFA0506000 to G.L.), the National Natural Science Foundation of China (81930108 to G.L., 21961142009 to G. L., 81670768 to Y.W., and 81670244 to W.H.), and Natural Science Foundation of Zhejiang Province (LR18H160003 to Y.W.).

## Author contributions

W.L., J.H., Y.J., Q.F., M.W., and Y.Z. performed the research. G.L. and Y.W. designed the research study. J.Q., X.C., and G.W. contributed essential reagents or tools. J.H., W.H., H. L., and G.L. analyzed the data. G.L., H.L., Z.K., and Y.W. wrote the paper.

## Competing interests

The authors declare no competing interests.
