## [Peer Review File · Nature Communications]

Reviewers' comments:

Reviewer #1 (Remarks to the Author):

Wang et al propose a novel mechanism underlying the HG-induced inflammatory myocardial injury of diabetic CMO using

both in-vitro and in-vivo studies. The key conclusions from the manuscript were that blockage of MD2 significantly inhibited pro-inflammatory TLR4-Myd88-NFkB/MAPKs signaling activation which correlated with reduced myocardial remodeling and improved cardiac function. Additionally, they identified that HG induced rapid generation of extracellular AGEs which eventually bind directly to MD2 to exert the signaling upon binding to its receptor.

In the first demonstration of TLR activation by HG, Dasu et al in Diabetes 2008, proposed increased ROS . Please exclude this mechanism.

Also what is the status and role of TLR2 which is also elevated with HG and in diabetes

In a subsequent paper in Diabetes Care 2010, Dasu et al showed increase in MD2 in T2DM patients and a correlation with the AGE, CML.

To better substantiate your thesis please use methyglyxol and CML modified BSA . Your AGE-BSA can have many contaminants including Endotoxin. Also your assay is not well validated.

Can you recapitulate your data with the addition of polymyxin B etc

Using MG or CML by HPLC to quantify your claimed rapid development of AGE will be more convincing

What is the mechanism of AGE engaging MD2-TLR4.

It is important to include patients with T2DM without CMO for your in-vivo study to further illustrate your hypothesis.

Please label MD2KO to better define the KO

Reviewer #2 (Remarks to the Author):

This is a very interesting study to characterize glucose-induced activation of TLR4 signaling in cardiac myocytes and monocytes and its implication in cardiac damage in diabetic condition. The authors show nicely that glucose and serum proteins are able to produce AGEs, which then bind directly to MD2, a co-receptor of TLR4 to trigger the receptor-induced and MyD88-mediated activation of NFkB to induce inflammatory response. Genetic deletion and pharmacological inhibition of MD2 significantly attenuate glucose and diabetes-induced cardiac inflammation and subsequently cardiac dysfunction and remodeling. Several concerns need to be addressed.

1. While authors have tried to rule out of a role of RAGE in glucose-induced activation of TLR4 signaling for expression of inflammatory factors such as TNF and IL6, there are abundant literatures supporting a role of RAGE in NFkB activation through phosphorylation of IKK, which is essential to inflammatory response. Perhaps both TLR4 and RAGE are necessary for glucose-induced inflammatory response? This reviewer has reservation on the observation in Supplementary Figure 13, in which knockdown of RAGE does not affect glucose-induced TNF and IL6 expression. More controls are necessary to dissect the relative contribution of both TLR4 and RAGE signaling systems in glucose-induced inflammatory response. A cleaner system with RAGE knockout may be helpful?

2. Another major concern is the discrepancy between STZ models and glucose studies described in the manuscript. First, depending on the genetic background, mice usually receive higher doses of STZ (160-240 mg/kg) than the described 100 mg/kg in this study to induce type 1 diabetes (<https://www.jax.org/jax-mice-and-services/find-and...mice/.../stz-induced-diabetes>; Lab Anim. 2011 Jul; 45(3): 131-140. and Comp Med. 2011 Aug; 61(4): 356-360). Second, on the page 6 line 121, the authors state that "the STZ mice have a fast glucose > 216 mg/dl by 7 days". This

statement (216 mg/dl is equivalent to 12 mmol/L) is not consistent with data in Supplemental Figure 2; and it is closer to those in control mice than STZ-injected mice (with 20 mmol/L and 25 mmol/L for WT and DM2 knockout mice, respectively). Lastly, the authors state in page 9 Line 195 that "Determination of a dose-response using glucose concentration range of 7-60 mM indicated that significant MD2-TLR4 complex formation occurred at >30 mM (Fig. 3B)". In most cellular studies, the glucose concentration is 33 mmol/L, which is equivalent to 600 mg/dl, two times higher than the glucose concentrations described in the STZ mice (Supplemental Figure 2). The authors need to show consistency between the in vivo and in vitro studies to support the conclusion that the observations in animal hearts are indeed due to the MD2/TLR4 signaling paradigm described in cellular studies.

3. The western blots of MD2 are of poor quality. For example, in Figure 1A and 1B, three different blots of MD2 show completely different patterns. It is not clear which band(s) are the right band(s) for MD2. The authors should also validate the antibody with MD2-KO tissues.

4. Please provide molecular weight markers on each gel. The overall quality of western blots is of concern, such as Figure 2C MMP, Figure 3D and 3E, Figure 6C and 6E, and Figure 7B and 7H. It would be essential that the authors provide images of high quality. The original gel documents would help evaluation.

Minor:

In the ITC assay in Figure legends of 4 and 6, the authors mislabeled "heat rate" as "heart rate"? Please proof read the manuscripts thoroughly.

Reviewer #3 (Remarks to the Author):

This interesting study by Wang et al describes the role of MD2 in the development of inflammatory diabetic cardiomyopathy. The authors show that knockout and pharmacological blockade of MD2 results in attenuation of heart tissue damage and conclude that high glucose indirectly activates MD2-TLR4 signaling/inflammation by producing higher AGEs-levels that bind to MD2. In general, the manuscript is well written and the data are novel and compelling, supporting the drawn conclusions. Shortcomings however remain such as vague information on which contribution MD2 has on the respective cell types involved and the mouse model used:

General/ major comments:

1. Mice receiving STZ showed a mean left ventricular ejection fraction of above 70%. While this represents a significant reduction compared to control animals this output capacity is still well in the normal range which raises the question of the relevance of this model.
2. Analysis in (global) KO and after administration of blocking compound (Figures 2 and 3) both do not allow differentiation between effects of MD2 on cardiac resident cells (cardiomyocytes, fibroblasts, endothelial cells) or accumulating leukocytes. Is MD2 equally important on all these cells in the context of hyperglycemia? This could be further investigated with the use of bone marrow transplantation/ reconstitution with WT or KO bone marrow.

Further comments:

1. Fig 1A: MD2 shows a double band (phosphorylation?) Which one was used for quantification? In case of phosphorylation: is this altered by STZ injection (or after AGEs stimulation later on?)
Minor: Representative WB image? MD2 appears to be completely absent 16w after STZ injection.
2. Is MD2 upregulated in human diabetes/DCM patients?
3. Fig. 1C: Do other cell types upregulate MD2 (endothelial cells, fibroblasts)? It's not clear why the author's focused on MD2 expression in cardiomyocytes/macrophages since cells negative for actin or F4/80 also appear to be MD2 positive

4. Fig 2: was this analysis also done at week 16? If so, this should be added to the figure legend.
5. Fig 2A: The authors state that the general morphology in WT mice is more disorganized. The representative images are not very convincing since the observed 'differences' may also result from cutting/mounting/processing the tissue sections. H&E staining may not be the best method to show disorganization.
6. Table 1: Representative M-Mode/B-Mode images would be a valuable addition.
7. Fig 2B-C: could the authors provide a quantification?
8. Fig 2H: Did the authors quantify? Normalization to unphosphorylated form of JNK/ERK etc might be more appropriate than normalization to GAPDH.
9. Is the recruitment/infiltration of macrophages altered in MD2 KO after DM induction? Are there other inflammatory cells involved e.g. neutrophils?
10. Fig 4E: The statistical analysis is not clear. Why did the authors compare HG no serum only to HG serum and not the ctrl? There should be a 4th group with ctrl serum/no serum. Is there a significant difference between HG and ctrl in the absence of serum (as already described in the literature)?
11. Fig. 4H: The experimental setup is not clear. Why are there two ctrl and two HG groups. How do they differ? Are the ctrl + HG groups to the left cultured initially with serum? The authors didn't observe differences between ctrl and HG (no serum) in fig. 4E. Why is there such a strong induction of cytokines now?
12. Fig 4J-L: There are studies showing that HG alone induces an inflammatory response (increased glycolysis). Was the FBS heat inactivated? If not: FBS can induce complement activation.
13. Fig. 5A: Here again could complement activation play a role? Is glycolysis affected which may lead to increased inflammation? The latter should be discussed.
14. The authors should use the SD instead of SEM (SEM describes the precision for an estimated population mean, whereas SD is a measure of data variability).
15. What supports the statement that the identified mechanism induces „hypertrophy“ (schematic drawing)? According to table 1 in comparison to control mice, no relevant thickening of the left ventricular or posterior walls are observed.
16. The description of the investigated human samples from patients with DCM is rather superficial. Were those patients with diabetes and concurrent ischemic heart disease or excluded CAD? Where samples collected at exacerbation of disease or during steady state? ... This should be further described.

Response to reviewer's comments

Manuscript: MD2 activation by direct AGE interaction drives inflammatory diabetic cardiomyopathy

Summary:

We would like to thank the reviewers for dedicating their time to review our work and for offering suggestions to improve it. We have performed additional studies, added new data figures and have significantly revised the text. Major points of emphasis in our revised manuscript include addressing the cellular source of MD2, contribution of glycolysis, potential role of RAGE, potential contaminants that may induce inflammatory responses, and providing a clearer mechanistic mode of AGE-MD2 interaction. We sincerely hope that the reviewers and the Editor find the manuscript compelling in the revised form.

Please find below, our point-by-point response to the reviewer's comments.

REVIEWER 1

Wang et al propose a novel mechanism underlying the HG-induced inflammatory myocardial injury of diabetic CMO using both in-vitro and in-vivo studies. The key conclusions from the manuscript were that blockage of MD2 significantly inhibited pro-inflammatory TLR4-Myd88-NFkB/MAPKs signaling activation which correlated with reduced myocardial remodeling and improved cardiac function. Additionally, they identified that HG induced rapid generation of extracellular AGEs which eventually bind directly to MD2 to exert the signaling upon binding to its receptor.

Reviewer #1-1: In the first demonstration of TLR activation by HG, Dasu et al in Diabetes 2008, proposed increased ROS. Please exclude this mechanism.

Response: Thank you for this suggestion. This is an important study and was mistakenly omitted from our citation list. Dasu and colleagues (PMID:18650365) showed increased protein levels of TLR4 and MyD88 in THP-1 monocytic cell line exposed to high levels of glucose. They also found that silencing P47 Phox (NADPH oxidase) prevents glucose-induced TLR4 expression. However, TLR4 activity was only shown to be inhibited by siRNA-targeted silencing of TLR4, and not upon attenuation of reactive oxygen species (ROS). It is certainly possible that ROS is involved in the upregulated expression of TLR4 and TLR4 adapter proteins, as reported by Dasu et al. Inhibition of TLR4 has also been shown to suppress NADPH oxidase and ROS levels (PMID: 21159162). Therefore, it appears that ROS and oxidative stress may prime target cells for TLR4 engagement by increasing its protein levels. We have revised the text to include this potential interaction but have kept the focus on our primary objective of identifying how TLR4 is activated by HG and the potential

role of MD2 in this process. We hope that this is acceptable to the reviewer.

Reviewer #1-2: Also what is the status and role of TLR2 which is also elevated with HG and in diabetes.

Response: Thanks. We have not assessed the status of TLR2 for a few important reasons. First, MD2 acts a co-receptor for TLR4 and not TLR2. Building on our recent discoveries of MD2 mediating the activities of oxidized-low-density lipoproteins (PMID: 31151053) and saturated fatty acids (PMID: 28045026), and landmark studies by other investigators firmly establishing the role of MD2 in TLR4 activation in innate immunity, we reasoned that MD2 may also play a critical role in mediating TLR4 activation and cardiac inflammation in diabetic cardiomyopathy. Second, even though studies show that both TLR4 and TLR2 are elevated in monocytes isolated from type 2 diabetic patients (PMID: 20067962), exposure of monocytes to high glucose levels lead to more pronounced effects on TLR4 compared to TLR2 (PMID: 18650365). Diabetic mice and rats also show increased TLR4 levels in heart tissues (PMID: 21159162, 25619390). Other target organs of diabetic complications such as kidneys also implicate TLR4 activity and not TLR2 (PMID: 22021706). Therefore, we focused our efforts on identifying the mechanisms by which TLR4 is activated and the utility of inhibiting MD2-TLR4 in diabetic cardiomyopathy.

Reviewer #1-3: In a subsequent paper in Diabetes Care 2010, Dasu et al showed increase in MD2 in T2DM patients and a correlation with the AGE, CML.

Response: Thank you for this comment. This study by Dasu and colleagues showing that TLR4 protein level and MD2 mRNA level are elevated in monocytes isolated from T2DM patients (PMID: 20067962) strengthens our rationale and conclusions. These elevated levels also correlated with CML. We have included this citation in our manuscript.

Reviewer #1-4: To better substantiate your thesis please use methyglyxol and CML modified BSA.

Response: Thank you for this suggestion. We have performed the suggested experiment. H9C2 cells were exposed to AGE-BSA or CML-BSA for 15 minutes and MD2-TLR4 interaction was assessed by co-immunoprecipitation. Our results show that CML-modified BSA does not induce MD2-TLR4 interaction beyond levels seen in control conditions. AGE-BSA, however, induced rapid MD2-TLR4 interaction. These results are not surprising since AGE products formed from different intermediates have differential activities in cells (PMID: 11016456). Glucose- and glycolaldehyde-BSA is significantly different compared to glyoxal- and methylglyoxal-BSA. Glucose- and glycolaldehyde-BSA share similar fluorescence patterns, low CML content, and uptake patterns in macrophages. We have incorporated this data in the new Supplementary Figure S21 and have revised the corresponding text.

Supplementary Figure S21: Effect of AGE- and CML-modified BSA on MD2-TLR4 interaction in H9c2 cells. H9C2 cells were exposed to 30 µg/mL AGE- or CML-modified BSA for 15 min. Proteins were immunoprecipitated using TLR4 antibody (IP) and levels of MD2 were determined by immunoblotting (IB) [$n = 3$].

Reviewer #1-5: Your AGE-BSA can have many contaminants including Endotoxin. Also your assay is not well validated.

Response: AGE-BSA used in our study was obtained from Biovision (cat no. 2221-10). It was prepared by reacting BSA with glycoaldehyde under sterile conditions. The endotoxin levels reported in datasheets for the lots used was less than 0.1 EU/µg. We have also measured endotoxin levels in our AGE-BSA using Chromogenic LAL Endpoint Assay Kit (Cat#: EC64405; Bioendo Technology Co., LTD, Xiamen, China). We found endotoxin levels less than 0.1 EU/µg of AGE-BSA, consistent with the reported levels from the vendor. We have added this information in the Methods section.

Reviewer #1-6: Can you recapitulate your data with the addition of polymyxin B etc

Response: This is an excellent suggestion. Although the AGE-BSA preparation used in our study contains very low endotoxin levels, we have performed the suggested study. Briefly, we exposed H9C2 cells to AGE-BSA with or without polymyxin B. Since polymyxin B binds to lipid A of LPS, any endotoxin contamination of AGE-BSA generating false-positive results would be mitigated. Consistent with the reported and tested levels of endotoxin (please see response to comment #1-5 above), presence of polymyxin B did not alter AGE-BSA induced inflammatory responses in H9C2 cells. These results suggest that inflammatory activation of cells upon AGE-BSA exposure is not attributed to endotoxin contamination. We have included the data in the new Supplementary Figure S23. Corresponding changes to the text have also been made.

Supplementary Figure S23: MD2-binding polymyxin B does not alter AGE-induced inflammatory responses in H9C2 cells. (A) H9C2 cells were pretreated with 30 $\mu\text{g}/\text{mL}$ polymyxin B for 1 hour and then exposed to 30 $\mu\text{g}/\text{mL}$ AGE-BSA (AGE). AGE-BSA (with or without polymyxin B pretreatment) were carried out for 15 minutes. Protein samples were immunoprecipitated using MD2 antibody (IP) and levels of TLR4 were determined by immunoblotting (IB). (B, C) H9C2 cells were exposed to 30 $\mu\text{g}/\text{mL}$ AGE-BSA for 24 hours, with or without pretreatment with 30 $\mu\text{g}/\text{mL}$ polymyxin B for 1 hour. Levels of TNF- α (B) and IL-6 (C) mRNA were measured. Data was normalized to β -actin [$n=4$; n.s. = no significance; $*p<0.05$].

Reviewer #1-7: Using MG or CML by HPLC to quantify your claimed rapid development of AGE will be more convincing.

Response: Thank you for this suggestion. We have measured MG and CML levels in H9C2 cells following exposure to high glucose (HG) as well as in heart tissues of diabetic mice. MG levels were detected using methods described previously (PMID: 30310608), and CML levels were measured by CircuLex CML/N ϵ -Carboxymethyl-lysine ELISA (cat no. CY-8066; Clinisciences, Nanterre, France). Our results show that HG exposure of H9C2 cells for up to 15 minutes does not increase CML levels and no changes were seen in MG levels following exposure to HG for up to 60 minutes (new Supplementary Figure S20). Furthermore, MG and CML were not increased in heart tissues of diabetic mice at 16-week followup (new Supplementary Figure S25B, S25C). These results were expected as intermediates and AGE products would depend on glucose levels and duration of glycation. It is quite possible that greater duration of H9C2 exposure to glucose and diabetes in mice may increase MG and CML levels.

Supplementary Figure S20: Levels of methylglyoxal and N ϵ -carboxymethyl-lysine in H9C2 cells exposed to high levels of glucose. (A, B) H9C2 cells were exposed to 33 mM glucose (HG) for different time periods, either in the absence (A) or presence of 10% heat-inactivated serum (B). Levels of methylglyoxal (MGO) were measured by two-photon fluorophore conjugated to o-phenylenediamine which contains a MGO recognition site [n = 3]. (C) H9C2 cells were exposed to 33 mM glucose for 15 minutes in the presence of 10% heat-inactivated serum and levels of CML were measured by ELISA [n = 3].

Supplementary Figure S25B-C: Levels of methylglyoxal and N ϵ -carboxymethyl-lysine in heart tissues of diabetic mice. (B, C) Levels of methylglyoxal (MGO, B) and N ϵ -carboxymethyl-lysine (CML, C) were measured in heart tissues of non-diabetic control mice and streptozotocin-induced diabetic mice. MGO levels were determined by a two-photon fluorophore conjugated to o-phenylenediamine as described in the Methods section. CML were measured by ELISA [n = 6].

Reviewer #1-8: What is the mechanism of AGE engaging MD2-TLR4.

Response: We have attempted to explore the mechanism by which AGE products engage MD2-TLR4 by carrying out two studies. First, we exposed H9C2 cells to high levels of glucose, either in the presence of functionally inactive LPS (Rhodobacter sphaeroides LPS, RS-LPS) or in its absence. RS-LPS binds to MD2 at the same site as LPS but does not initiate MD2-mediated TLR4 signaling. Our results show that RS-LPS is able to reduce HG-induced MD2-TLR4 interaction (Supplementary Figure S29A). However, it was not a complete normalization, suggesting that glucose-induced AGE products share some, but not all, LPS-binding sites on MD2. In support of this idea, we show that exposure of H9C2 cells to high glucose levels displaces cell surface LPS binding (Supplementary Figure S29B). Together, these results suggest that glucose-induced AGE products may have overlapping binding interaction with MD2. However, as stated in our manuscript, delineating the exact mode and sites of AGE products would be very challenging due to the heterogenous nature of AGE products. Finally, an AGE-MD2-TLR4 co-crystal structure may be needed for the clear illustration of AGE engaging MD2-TLR4 complex. We have revised the Discussion section to address the potential mode of interaction.

Reviewer #1-9: It is important to include patients with T2DM without CMO for your in-vivo study to further illustrate your hypothesis.

Response: We fully agree with the reviewer that diabetic patients without cardiomyopathy would add to the overall idea and conclusions. However, as the reviewer may appreciate,

obtaining these samples is very difficult. We were only able to collect blood samples from 8 non-diabetic subjects and 9 diabetic patients with cardiomyopathy for peripheral blood mononuclear cell isolation and serum preparation. More difficultly, we were only able to obtain 7 serum samples from diabetic subjects without cardiomyopathy. These serum samples were used to measure MD2, TNF- α , and IL-6 levels. We have included the new data in the following figure, which showed that the serum MD2, TNF- α , and IL-6 levels in diabetic subjects with cardiomyopathy is significantly higher than that in normal subjects but lower than that in DCM patients. Since we have only serum samples from diabetic subjects without cardiomyopathy, we did not show the serum MD2, TNF- α , and IL-6 levels in the revised Supplementary Figure S27C-S27E. We think that the current data are able to support our main conclusion. We have also addressed this limitation in our Discussion section.

Levels of serum MD2 and cytokines in human diabetic subjects: Serum samples were obtained from healthy volunteers (Con, non-diabetic control), diabetic subjects with cardiomyopathy (DCM), and diabetic subjects without cardiomyopathy (D -CM). Levels MD2 (C), TNF- α (D), and IL-6 (E) in human serum samples [Mean \pm SEM; n = 7; *p<0.05].

Reviewer #1-10: Please label MD2KO to better define the KO

Response: Thank you for pointing this out. We have revised the figures and text to change MD2 to MD2KO.

REVIEWER 2

This is a very interesting study to characterize glucose-induced activation of TLR4 signaling in cardiac myocytes and monocytes and its implication in cardiac damage in diabetic condition. The authors show nicely that glucose and serum proteins are able to produce AGEs, which then bind directly to MD2, a co-receptor of TLR4 to trigger the receptor-induced and MyD88-mediated activation of NFκB to induce inflammatory response. Genetic deletion and pharmacological inhibition of MD2 significantly attenuate glucose and diabetes-induced cardiac inflammation and subsequently cardiac dysfunction and remodeling. Several concerns need to be addressed.

Reviewer #2-1: While authors have tried to rule out of a role of RAGE in glucose-induced activation of TLR4 signaling for expression of inflammatory factors such as TNF and IL6, there are abundant literatures supporting a role of RAGE in NFκB activation through phosphorylation of IKK, which is essential to inflammatory response. Perhaps both TLR4 and RAGE are necessary for glucose-induced inflammatory response? This reviewer has reservation on the observation in Supplementary Figure 13, in which knockdown of RAGE does not affect glucose-induced TNF and IL6 expression. More controls are necessary to dissect the relative contribution of both TLR4 and RAGE signaling systems in glucose-induced inflammatory response. A cleaner system with RAGE knockout may be helpful?

Response: We thank the reviewer for pointing this surprising finding out and suggesting we perform additional studies to confirm. We certainly were puzzled by the finding, as we expected RAGE knockdown to have some effect on cytokine production, at least when exposed to AGE-BSA. We initially thought that there were two potential reasons as to why we did not observe changes in inflammatory cytokine production from HG or AGE-BSA exposure. First, the knockdown efficiency may have not been adequate to see any significant changes, as implied by the reviewer. Although our siRNA transfections reduced the amount of RAGE proteins to 50% the control levels, it is quite possible that this level of knockdown is not sufficient. Second possibility was that RAGE may require TLR4 for its downstream signaling. This possibility is supported by recent studies by Valencia and colleagues who showed that certain AGE products fail to bind RAGE and upregulate VCAM-1 in endothelial cells unless endotoxin was added (PMID: 15127201). However, we know that our AGE-BSA solutions have almost undetectable levels of endotoxin (tested in the laboratory). Therefore, as suggested by the reviewer, we have performed additional experiments to address the first possibility that RAGE knockdown level was not enough. We transfected H9C2 cells with plasmids encoding RAGE shRNA. This allowed us to silence approximately 90% of RAGE expression, as determined by immunoblotting. These new experiments with significant RAGE silencing, presented in the revised Supplementary Figure S24, show two important results: 1) RAGE knockdown does not alter HG- and AGE-induced MD2-TLR4 complex formation, and 2) RAGE knockdown suppresses HG- and AGE-induced inflammatory cytokine production but does not completely normalize it. Therefore, it appears that RAGE and MD2-TLR4 represent two distinct pathways that induce HG- and AGE-mediated inflammatory responses in cells. We have included the new data and have revised the corresponding text in the manuscript.

Revised Supplementary Figure S24: RAGE knockdown does not alter HG- or AGEs-induced MD2-TLR4 activation in H9C2 cells. (A) RAGE in H9C2 cells was knocked down by shRNA (sh-RAGE) transfection. Control cells were transfected with negative control (NC) plasmids. Representative Western blot showing RAGE protein levels following transfection. GAPDH was used as loading control [n=3; Con = untransfected cells, NC = negative control plasmid, Sh-RAGE = RAGE shRNA plasmid]. The column figure shows a silence approximately 90% of RAGE expression. (B) H9C2 cells transfected with RAGE shRNA were exposed to HG (33 mM glucose) for 5 min. Proteins were immunoprecipitated using TLR4 antibody (IP) and levels of MD2 were determined by immunoblotting (IB) [n=3]. (C, D) mRNA levels of TNF α (C) and IL-6 (D) in H9C2 cells were determined following RAGE shRNA transfection and exposure to HG (33 mM glucose). HG exposure was carried out for 6 h [Mean \pm SEM; n=3; *p<0.05, **p<0.01, ***p<0.001]. (E) RAGE shRNA transfected H9C2 cells were exposed to HG (33 mM glucose) for 5 min. Interactions between AGE-MD2 and AGE-TLR4 were determined by co-immunoprecipitation [IP = precipitating antibody, IB = immunoblot antibody; n=3]. (F, G) H9C2 cells transfected with RAGE shRNA were exposed to 33 μ g/mL AGE-BSA (AGE) for 6 h. mRNA levels of TNF- α (F) and IL-6 (G) were measured [Mean \pm SEM; n=3; *p<0.05, **p<0.01, ***p<0.001].

Reviewer #2-2: Another major concern is the discrepancy between STZ models and glucose studies described in the manuscript. First, depending on the genetic background, mice usually receive higher doses of STZ (160-240 mg/kg) than the described 100 mg/kg in this study to induce type 1 diabetes (<https://www.jax.org/jax-mice-and-services/find-and...mice/.../stz-induced-diabetes>; Lab Anim. 2011 Jul; 45(3): 131–140. and Comp Med. 2011 Aug; 61(4): 356–360). Second, on the page 6 line 121, the authors state that “the STZ mice have a fast glucose > 216 mg/dl by 7 days”. This statement (216 mg/dl is equivalent to 12 mmol/L) is not consistent with data in Supplemental Figure 2; and it is closer to those in control mice than STZ-injected mice (with 20 mmol/L and 25 mmol/L for WT and DM2 knockout mice, respectively). Lastly, the authors state in page 9 Line 195 that “Determination of a dose-response using glucose concentration range of 7-60 mM indicated that significant MD2-TLR4 complex formation occurred at >30 mM (Fig. 3B)”. In most cellular studies, the glucose concentration is 33 mmol/L, which is equivalent to 600 mg/dl, two times higher than the glucose concentrations described in the STZ mice (Supplemental Figure 2). The authors need to show consistency between the *in vivo* and *in vitro* studies to support the conclusion that the observations in animal hearts are indeed due to the MD2/TLR4 signaling paradigm described in cellular studies.

Response: Thank you for this comment. Please allow us to address the three sections of the comments: 1) dose of STZ for the *in vivo* studies, 2) blood glucose levels for our designation of diabetic mice, 3) association of glucose levels in culture studies and experimental animals.

1) The reviewer is absolutely correct that different investigators have used different doses of streptozotocin to induce diabetes in experimental models. We have previously used 100 mg/kg single STZ injection in C57BL/6 to induce diabetes (PMID: 25736300, 25758431). This dose is consistent with studies showing that C57BL/6 mice are among the high responders to STZ (PMID: 16118394). In fact, Hayashi and colleagues reported that the smallest single dose of STZ required to induce diabetes in male C57BL/6 mice is 100 mg/kg (PMID: 16755002). In contrast, immunodeficient mice (nude and BALB/c) require 150-160 mg/kg (PMID: 16755002, 22330251). We certainly believe that we cannot assume that mice will react to STZ in the same manner. Therefore, it is imperative that mice are tested and allocated to control/non-diabetic and diabetic groups appropriately, which is the second part of the reviewer’s comment.

2) We apologize for the incorrect and confusing statement regarding blood glucose levels following STZ injection. We measured fasting blood glucose levels in mice following STZ injection. Only mice showing 12 mM fasting glucose levels (216 mg/dL) on 3 consecutive days were considered diabetic. Once the mice were divided into non-diabetic control and diabetic groups, their blood glucose levels and body weights were measured on a weekly basis. The reviewer is correct that by 7 days post-STZ injection, blood glucose levels were recorded at approximately 20 mM. Our designation of diabetic state is based on previous studies showing 200 mg/dL glucose levels (11.1 mM) on three consecutive days in mice following single STZ injection (PMID: 9532197). We hope that this clarification in the manuscript satisfies the reviewer and the readers.

3) We thank the reviewer for suggesting that we show consistency between the *in vivo* and *in vitro* studies. However, we believe that correlating/linking levels of glucose in cultured cells and experimental animals is very problematic. As the reviewer knows, both cultured

cells and animals are used as experimental models to understand the observations made in clinical samples and in patients. These experimental models are primarily based on the stimulus (glucose) generating a readout. Almost all studies conducted in cultured cells are acute studies and performed in a purified system. Therefore, the concentration of stimulus in cultured studies is almost always different compared to levels found *in vivo*. Our *in vitro* studies were performed by exposing H9C2 cells and macrophages to 33 mM glucose. This concentration is based on our previous studies (PMID: 25736300, 25758431) as well as studies conducted by other groups (PMID: 23884890, 12933701). We are confident in our conclusions that high levels of glucose induce inflammatory responses in cardiomyocytes and macrophages through AGE generation and MD2-mediated TLR4 activation, as MD2 silencing and inhibition by L6H21 prevented inflammatory cytokine production. In addition, mannitol used at the same concentration did not induce inflammatory responses (Supplementary Figure S10). Similarly, MD2 deficiency in mice or inhibition prevented diabetes-induced inflammatory cytokines and tissue damage. Therefore, we believe that the *in vitro* and *in vivo* experimental modeling, despite the different levels of glucose, show MD2/TLR4 activation.

Reviewer #2-3: The western blots of MD2 are of poor quality. For example, in Figure 1A and 1B, three different blots of MD2 show completely different patterns. It is not clear which band(s) are the right band(s) for MD2. The authors should also validate the antibody with MD2-KO tissues.

Response: Thank you very much for pointing this out. We have replaced the figures with higher-quality blot images. Also, please see our response to Reviewer #3-3.

Reviewer #2-4: Please provide molecular weight markers on each gel. The overall quality of western blots is of concern, such as Figure 2C MMP, Figure 3D and 3E, Figure 6C and 6E, and Figure 7B and 7H. It would be essential that the authors provide images of high quality. The original gel documents would help evaluation.

Response: Thank you. We have revised the images in these Figures and have included band size information, as suggested by the reviewer.

Reviewer #2-5 (Minor): In the ITC assay in Figure legends of 4 and 6, the authors mislabeled “heat rate” as “heart rate”? Please proof read the manuscripts thoroughly.

Response: Thank you. We have corrected the figures and figure legends. We have also proofread the manuscript to correct mistakes.

REVIEWER 3

This interesting study by Wang et al describes the role of MD2 in the development of inflammatory diabetic cardiomyopathy. The authors show that knockout and pharmacological blockade of MD2 results in attenuation of heart tissue damage and conclude that high glucose indirectly activates MD2-TLR4 signaling/inflammation by producing higher AGEs-levels that bind to MD2. In general, the manuscript is well written and the data are novel and compelling, supporting the drawn conclusions. Shortcomings however remain such as vague information on which contribution MD2 has on the respective cell types involved and the mouse model used:

Reviewer #3-1: Mice receiving STZ showed a mean left ventricular ejection fraction of above 70%. While this represents a significant reduction compared to control animals this output capacity is still well in the normal range which raises the question of the relevance of this model.

Response: Unfortunately, this is a known limitation of the relatively short-term STZ model and indicative of heart failure with preserved ejection fraction. Relative to non-diabetic controls, STZ-induced diabetes does show significantly depressed ejection fraction, as reported in our study as well as by others (PMID: 29432575, 21415098, 27621594). We agree with the reviewer that in isolation, depressed ejection fraction which is still in the normal range raises questions regarding the model. However, we and others always consider ejection fraction decreases with other pathological indices of cardiac damage. Our results of increased creatine kinase-muscle/brain, and cardiac fibrosis and hypertrophy indicate significant overall cardiac dysfunction.

Reviewer #3-2: Analysis in (global) KO and after administration of blocking compound (Figures 2 and 3) both do not allow differentiation between effects of MD2 on cardiac resident cells (cardiomyocytes, fibroblasts, endothelial cells) or accumulating leukocytes. Is MD2 equally important on all these cells in the context of hyperglycemia? This could be further investigated with the use of bone marrow transplantation/ reconstitution with WT or KO bone marrow.

Response: This is an excellent suggestion. We have performed two extensive studies to offer insight into the contribution of different cardiac cells to MD2. First, we isolated cardiac fibroblasts and endothelial cells from B6 mice. The levels of MD2 mRNA and protein were measured in these cells and compared to primary cardiomyocytes and primary macrophages. We show that both endothelial cells and fibroblasts do express MD2, but the levels are significantly lower compared to cardiac myocytes and macrophages. These results are included in the new Supplementary Figure S8. Secondly, we reconstituted irradiated MD2 knockout mice with bone marrow cells from either wildtype or MD2 knockout mice. Three groups of mice (control B6, irradiated MD2 knockout mice reconstituted with wildtype B6 marrow cells, and irradiated MD2 knockout mice reconstituted with MD2 knockout marrow cells) were then made diabetic by 100 mg/kg STZ. A 12-week follow-up showed significantly reduced fibrosis (new Supplementary Figure S9A) and expression of hypertrophic marker ANP, fibrogenic factors collagen 1 and TGF- β , as well as reduced MMP2 levels (new

Supplementary Figure S9B). These results suggest that cardiomyocytes and macrophages play a critical role in MD2-mediated cardiac dysfunction. The results are also consistent with our recent studies showing that high fat-diet induced deleterious cardiac remodeling is mediated primarily through MD2 in cardiomyocyte and macrophages (*Nature Communications*, 2017, 8: 13997).

Supplementary Figure S8: MD2 expression in cardiac cells. MD2 expression level in cardiac cells was determined by qPCR (A) and immunoblotting (B). Primary cardiomyocytes, endothelial cells, and fibroblasts were isolated from heart tissues of C57BL/6 mice. Mouse peritoneal macrophages (MPMs) were also analyzed. Heart tissue RNA and protein lysates were used as control. mRNA levels were normalized to β -actin [Mean \pm SEM; n = 4; ns = not significant; *p<0.05]. GAPDH was used as loading control for immunoblotting.

Supplementary Figure S9: Bone marrow reconstitution reveals important contribution of cardiac cell- and macrophage-derived MD2 in diabetes-induced cardiac injury. MD2^{-/-} mice (MD2KO) were given

acidified water containing neomycin and polymyxin B sulphate. One day prior to transplantation, mice were subjected to total body irradiation (6 Gy). Bone marrow cells prepared from wildtype (WT) or MD2KO mice were prepared and injected intravenously at 5×10^6 cells. Tail clip samples and peritoneal macrophages were used for genotyping and confirmation of reconstitution. Mice were then made diabetic by streptozotocin (STZ). Heart tissues were harvested 12 weeks following the onset of diabetes. (A) Representative staining images of heart tissues showing H&E (upper panel), Sirius Red (middle panel), and Masson's Trichrome (lower panel) [n = 5]. (B) mRNA levels of ANP, Collagen-1, MMP2, and TGF- β in heart tissues as determined by qPCR. mRNA data was normalized to β -actin [Mean \pm SEM; n = 5 per group; ns = not significant; *p<0.05].

Reviewer #3-3 (Further comments: 1): Fig 1A: MD2 shows a double band (phosphorylation?) Which one was used for quantification? In case of phosphorylation: is this altered by STZ injection (or after AGEs stimulation later on?) Minor: Representative WB image? MD2 appears to be completely absent 16w after STZ injection.

Response: Unfortunately, the second band in the original Figure 1A is a not a phosphorylated MD2 form. We have only occasionally seen the double band and it most likely a result of non-specific binding. As noted in response to Reviewer #2-4, we have changed the figure to a higher-quality blot image.

Reviewer #3-4 (Further comments: 2): Is MD2 upregulated in human diabetes/DCM patients?

Response: We have performed additional analyses of samples obtained from diabetic patients. The new results show increased levels of MD2 in serum samples of diabetic patients, with or without evidence of cardiomyopathy. These new data have been incorporated in the revised manuscript. Please see our response to Reviewer #1-9, and revised Supplementary Figure S27B and S27C.

Reviewer #3-5 (Further comments: 3): Fig. 1C: Do other cell types upregulate MD2 (endothelial cells, fibroblasts)? It's not clear why the author's focused on MD2 expression in cardiomyocytes/macrophages since cells negative for actin or F4/80 also appear to be MD2 positive.

Response: This is indeed an interesting question. Our immunostaining of heart tissues shows cardiomyocytes and macrophages as the major cell types expressing MD2. However, other cells were also slightly positive for MD2 immunoreactivity. As indicated in response to Reviewer #3-2 above, other cardiac cells such as endothelial cells and fibroblasts also express MD2 but the levels are significantly lower compared to cardiomyocytes and macrophages. Therefore, there may be some contribution to inflammatory cytokines in heart tissues from endothelial cells and possibly fibroblasts. We have included this limitation in our discussion.

Reviewer #3-6 (Further comments: 4): Fig 2: was this analysis also done at week 16? If so, this should be added to the figure legend.

Response: We have revised the legend to clearly state the end-point. Thank you for the comment.

Reviewer #3-7 (Further comments: 5): Fig 2A: The authors state that the general morphology in WT mice is more disorganized. The representative images are not very convincing since the observed 'differences' may also result from cutting/mounting/processing the tissue sections. H&E staining may not be the best method to show disorganization.

Response: We included H&E and basic description of observations on the structure as this is the preferred data for readers with Pathology expertise. We have revised the language accordingly and have described the changes we see as purely descriptive. We do feel that H&E images add value to some readers and have elected to keep the data in the manuscript. We hope that is acceptable to the reviewer.

Reviewer #3-8 (Further comments: 6): Table 1: Representative M-Mode/B-Mode images would be a valuable addition.

Response: We agree with the reviewer that M-Mode/B-Mode images would provide a valuable addition. However, we are sorry that these traces were not obtained. Unless we carry out the whole experiment again, we will not be able to add these at this time.

Reviewer #3-9 (Further comments: 7): Fig 2B-C: could the authors provide a quantification?

Response: Thank you for the suggestion. We have provided the quantification in the new Supplementary Figure S3 and S4A.

Reviewer #3-10 (Further comments: 8): Fig 2H: Did the authors quantify? Normalization to unphosphorylated form of JNK/ERK etc might be more appropriate than normalization to GAPDH.

Response: We have provided the quantification in Supplementary Figure S5. The data of phosphorylated proteins was normalized GAPDH. We have previously shown no changes in total protein levels of MAPK (ERK, JNK, and p38) in heart tissues of MD2 knockout mice (PMID: 28013347, 28045026, 28965884) and cultured cells transfected with MD2 siRNA or inhibited by L6H21 (PMID: 28013347, 28965884).

Reviewer #3-11 (Further comments: 9): Is the recruitment/infiltration of macrophages altered in MD2 KO after DM induction? Are there other inflammatory cells involved e.g. neutrophils?

Response: Thank you for this comment. We have performed an additional analysis by staining tissues with CD68 antibody to detect macrophages. The data has been added in the new Supplementary Figure S6. Results show reduced CD68 immunoreactivity in heart tissues of MD2 knockout mice, suggesting decreased infiltration. These findings are consistent with other studies showing reduced macrophage infiltration in the context of TLR4 deficiency

(PMID: 29393431, 30851101). Myocardial inflammation is a sum effect of cardiac myocytes expressing various pro-inflammatory cytokines, cell surface molecules, and chemokines that recruit macrophages. Recruited cells substantially increase inflammatory factors in the heart. Our studies with marrow reconstitution of irradiated mice shows that both cardiac MD2 and specific hematopoietic cell MD2 contributes to cardiac fibrosis and dysfunction. We have incorporated the new data and elaborated on the findings in the revised manuscript.

We have not assessed neutrophil infiltration in our model system. Neutrophils are believed to be short-lived effector cells. However, they play an important role in setting the stage for macrophage recruitment and polarization. Since our model system required 16 weeks of diabetes, we have focused mostly on the persistent leukocyte subset (macrophage). It is possible that myeloperoxidase activity is still elevated at this time point, however, it may still not provide a reliable assessment of neutrophil numbers. If the review insists that different leukocyte subsets should be examined, we will do so in diabetic mouse hearts.

Supplementary Figure S6: MD2 deficiency prevents macrophage infiltration in hearts of diabetic mice.

(A) Representative immunohistochemical staining of heart tissues for macrophage marker CD68. Heart tissues were harvested from mice at 16 weeks following the onset of streptozotocin (STZ)-induced diabetes. Immunoreactivity was detected by diaminobenzidine (brown) [WT = wildtype, Con = nondiabetic controls, STZ = streptozotocin-induced diabetic mice, MD2KO = MD2^{-/-} mice]. (B) Lysates prepared heart tissues were subjected to immunoblotting for macrophage marker CD68, and adhesion molecules intercellular cell adhesion molecule-1 (ICAM-1) and vascular cell adhesion molecule (VCAM-1). GAPDH was used as loading control [n = 6 or 7 per group].

Reviewer #3-12 (Further comments: 10): Fig 4E: The statistical analysis is not clear. Why did the authors compare HG no serum only to HG serum and not the ctrl? There should be a 4th group with ctrl serum/no serum. Is there a significant difference between HG and ctrl in the absence of serum (as already described in the literature)?

Response: Thanks. There is no significant difference between the HG no serum group and the Ctrl group in Figure 4E. Here, since we plan to show no serum significantly inhibited HG-induced inflammation, we only compared HG no serum group to HG serum. Please see Figure 4G, 4H, 4J, and 4K, where we compared HG and control in the absence of serum.

Reviewer #3-13 (Further comments: 11): Fig. 4H: The experimental setup is not clear. Why are there two ctrl and two HG groups. How do they differ? Are the ctrl + HG groups to the left

cultured initially with serum? The authors didn't observe differences between ctrl and HG (no serum) in fig. 4E. Why is there such a strong induction of cytokines now?

Response: We apologize for the lack of clarity in the experimental design. One set of control and HG was performed in the absence of serum and one set was performed in its presence. We have corrected the bar above the two sets, revised the figure legend, and revised the statistical analyses. We hope that the design is more logical with these corrections.

Reviewer #3-14 and #3-15 (Further comments: 12 and 13): (#3-14; comment 12) Fig 4J-L: There are studies showing that HG alone induces an inflammatory response (increased glycolysis). Was the FBS heat inactivated? If not: FBS can induce complement activation. **(#3-15; comment 13)** Fig. 5A: Here again could complement activation play a role? Is glycolysis affected which may lead to increased inflammation? The latter should be discussed.

Response: Thank you for these comments. Our FBS was heat-inactivated before performing the experiments. We have included this detail in the Methods section. We have also performed an additional experiment to test whether FBS which has not been heat-inactivated would generate false-positive data through complement activation. We cultured H9C2 cells for 10 days in the presence of FBS which was heat-inactivated and FBS which did not go through heat-inactivation. Cells were then exposed to 33 mM glucose. In both FBS conditions, induction of TNF- α and IL-6 was similar. Therefore, we do not believe that the readout is attributed to complement activation. We have included this new data in our revised manuscript as Supplementary Figure S17.

Supplementary Figure S17: Effect of heat-inactivation on HG-induced inflammatory cytokine production in H9C2 cells. (A, B) H9C2 cells were cultured in media containing 10% heat-inactivated fetal bovine serum or serum that did not undergo heat-inactivation for 10 days. Cells were then exposed to HG (33mM glucose) for 24 h. mRNA levels of TNF- α (A) and IL-6 (B) were determined by real-time qPCR assay [Mean \pm SEM; n = 3; n.s. = not significant; ***p < 0.001 compared to respective controls].

In terms of glycolysis leading to inflammatory signaling, we have measured glycolytic activity in H9C2 cells using Seahorse XF Glycolysis Stress Test Kit. We exposed H9C2 cells

and MPMs to 33 mM glucose and noted that glycolysis inhibitor 2-DG changed glycolysis in H9c2 cells (new Supplementary Figure S18). To rule out glycolysis in inducing MD2-TLR4 signaling activation, we pretreated H9C2 cells with 2-DG and then exposed the cells to 33mM glucose. Co-immunoprecipitation showed that 2-DG does not alter HG-induced MD2-TLR4 interaction in H9C2 cells (Supplementary Fig. S19A). Similarly, of glycolysis inhibitor 2-DG dose not suppress HG-increased mRNA levels of TNF- α and IL-6 in H9c2 cells (Supplementary Fig. S19B-C). In contrast, glycolysis inhibitor 2-DG aggravated HG-induced TNF- α and IL-6 expression, possibly due to the decreased glucose consuming and increased extracellular glucose concentration. These results suggest that in H9C2 cells, MD2-TLR4 complex formation and inflammatory responses to HG are not mediated through increased glycolysis.

Supplementary Figure S18: Effect of high glucose on glycolytic function of cardiomyocytes and macrophages. Glycolytic function of cells was determined by Agilent Seahorse XF Glycolysis Stress Test Kit. Extracellular acidification rate (ECAR; mpH/min/ μ g protein) in H9C2 (A, 88000 cells per group) and MPMs (B, 22000 cells per group) exposed to HG (33 mM glucose). Sequential addition of glucose (Glu) and 2-deoxy-glucose (2-DG) is indicated by arrows. No changes in glycolysis was observed in H9C2 cells exposed to HG compared to cells cultured in 5 mM glucose [$n = 3$ or 4]. Glycolysis was increased in MPMs exposed to HG [$n = 3$ or 4].

Supplementary Figure S19: Effect of glycolysis inhibitor on HG-induced MD2-TLR4 complex formation and inflammation. (A) H9C2 cells were pretreated with 50 mM 2-DG for 30 min and then exposed to HG (33mM glucose) for 15 min. Proteins were immunoprecipitated with TLR4 antibody (IP) and levels of MD2 were determined by immunoblotting (IB) [$n = 3$]. (B and C) H9C2 cells were pretreated with 50 mM 2-DG for 30 min and then exposed to HG (33mM glucose) for 12h. mRNA levels of TNF- α (B) and IL-6 (C) were determined by real-time qPCR assay. Glycolysis inhibitor 2-DG dose not suppress HG-increased mRNA levels of TNF- α and IL-6 in H9c2 cells. In contrast, glycolysis inhibitor 2-DG aggravates HG-induced TNF- α and IL-6 expression, possibly due to the decreased glucose consuming and increased extracellular

glucose concentration. [Mean \pm SEM; n = 3; ns = not significant; *p<0.05].

Reviewer #3-16 (Further comments: 14): The authors should use the SD instead of SEM (SEM describes the precision for an estimated population mean, whereas SD is a measure of data variability).

Response: Although SEM and SD are related, we have no intention of misleading the reader into underestimating the variability in our data. Our approach has always been to clearly indicate SEM in all of our figures and perform proper statistical analysis. If, however, the reviewer feels that all graphs should be revised to Mean \pm SD, we would be willing to do so.

Reviewer #3-17 (Further comments: 15): What supports the statement that the identified mechanism induces „hypertrophy“(schematic drawing)? According to table 1 in comparison to control mice, no relevant thickening of the left ventricular or posterior walls are observed.

Response: Hypertrophy in the schematic was included based on our observation that that diabetes in mice increases cardiomyocyte size (Supplementary Figure S2C) and levels of hypertrophy marker ANP (Supplementary Figure S3, S7E). In addition, high glucose exposure of H9C2 cells increased β -MyHC levels (Supplementary Figure S15). In both systems, deficiency in MD2 prevented these changes. Although not directly tested in our study, AGE administration has been shown to mediate cardiac hypertrophy (PMID: 28901418). Therefore, we included hypertrophy in the schematic. If, however, the reviewer feels strongly that experimental evidence linking hypertrophy to the identified mechanisms is not strong, we will be willing to revise the figure.

Reviewer #3-18 (Further comments: 16): The description of the investigated human samples from patients with DCM is rather superficial. Were those patients with diabetes and concurrent ischemic heart disease or excluded CAD? Where samples collected at exacerbation of disease or during steady state? ... This should be further described.

Response: Thank you for the comment. We fully agree with the reviewer that this is a limitation of our study. It is quite difficult to obtain patient samples. Our inclusion criteria included diabetic patients with reduced EF%, and without the indication of other cardiac diseases. We have clarified the description of the human samples with diabetic cardiomyopathy in the Methods and Discussion section.

Reviewers' comments:

Reviewer #1 (Remarks to the Author):

This is a very unsatisfactory response to my review
The data on ROS induction of TLRs have been shown by many groups
Your rebuttal is unsatisfactory
Convincing data show a role of TLR2 in diabetic Nephropathy see Devaraj et al ATVB , etc - you cannot ignore published data to fit your hypothesis
Diabetes spans many decades what is the need to show an effect of 15 minutes of HG. What is the clinical relevance ?
I am not convinced what is your AGE moiety
You should repeat experiments with AGE that has MG or CML etc
Otherwise your data is troubling
If MG and CML does not cross react in your assay I still maintain (which you ignored 1-5) that your assay is unacceptable and not validated for mainstream research.
I need to see MD2 in mononuclear Cells as protein in T2DM patients without CMO
I am unclear what is the relevance of circulating MD2

Reviewer #2 (Remarks to the Author):

In this revised manuscript, the authors have done a thorough job to address all the concerns raised with extensive amount of new data. The manuscript has been significantly improved. I have only one minor comment. The author should list heart weights, and heart weight over body weight ratios in the paper since this is a critical perimeter for cardiac remodeling in diabetic cardiomyopathy.

Reviewer #3 (Remarks to the Author):

The authors have taken great effort to revise their manuscript on the role of MD2 in the development of inflammatory diabetic cardiomyopathy and have improved their work. While the presented additional mechanistic data are compelling, I have to admit that I am still not convinced that there is really a clinical relevance to the model used (see my previous Q1). The given answer that "other pathological indices of cardiac damage are also taken into consideration" does not alter the fact that an ejection fraction of the left ventricle above 70% is far from any relevant impairment that could lead to heart failure (symptoms). If the argument is taken that diastolic heart function might play a role, then stringent further analysis would be required (see e.g. doi: 10.1038/s41586-019-1100-z). Another aspect that is irritating is the response to my previous Q8 that "M-Mode/B-Mode ... traces were not obtained". This appears to be rather odd since these traces/images had to be acquired for the analysis in the first place.

Response to reviewer's comments (R2)

Manuscript: MD2 activation by direct AGE interaction drives inflammatory diabetic cardiomyopathy.

Summary: We thank the three reviewers for assessing the revisions to our study. We offer our sincere apologies if some of our responses were not adequate. We have performed additional experiments to illustrate the potential involvement of ROS and TLR2 (Reviewer 1), and to measure inflammatory factors in primary macrophages in response to methylglyoxal and CML (Reviewer 1). We have also added heart weights and heart weight to body weight ratios (Reviewer 2), and additional parameters of cardiac function to support the animal model of cardiac dysfunction (Reviewer 3). Lastly, we have included the M-mode images (Reviewer 3) and addressed other clarification issues that reviewers raised.

Please find below, our point-by-point response to the reviewer's comments.

REVIEWER 1

Reviewer #1-1: This is a very unsatisfactory response to my review.

Response: We apologize that we were unable to address the concerns raised by the reviewer adequately. In the first revision, we performed experiments with CML-BSA to show that MD2-TLR4 activation may be specific to glucose- and glycolaldehyde-BSA AGE-BSA, measured endotoxin levels in our AGE-BSA preparations and confirmed the results using polymyxin B, measured MGO and CML in cells and heart tissues of mice using commercially available kits and published techniques, and measured inflammatory cytokines in serum samples from diabetic patients. We do acknowledge that issues concerning TLR2 and the potential involvement of ROS in TLR expression were only addressed in response to the reviewer and not by performing additional experiments, for reasons outlined previously. In this second revision, we have performed studies to investigate the potential involvement of TLR2 and ROS, and additional assays to investigate CML and MGO. We sincerely hope that this second round of revisions addresses the reviewer's concerns.

Reviewer #1-2: The data on ROS induction of TLRs have been shown by many groups. Your rebuttal is unsatisfactory.

Response: We agree and indicated that ROS has been shown to induce TLR expression. Specifically, the study cited by the reviewer in the first revision clearly shows that that ROS mediates HG-induced expression of TLR2\TLR4 in a monocytic cell line (PMID: 18650365). We have performed additional experiments to measure TLR2\TLR4 expression in primary mouse macrophages and H9C2 cells exposed to high glucose (HG), and the potential involvement of ROS in such an induction. Our results (shown below and included as new *Supplementary Figure S17*) show that a 12-hour challenge of cells with HG increases the mRNA level of TLR4 in macrophages and H9C2 cells. Increases in mRNA levels of inflammatory cytokines (TNF- α and IL-6) were also observed in HG-exposed cells.

Pretreatment of cells with a ROS scavenger N-acetylcysteine (NAC) significantly reduced HG-induced levels of TLR4 and inflammatory factors but did not completely normalize these alterations. In addition, interestingly, NAC did not reduce the interaction between MD2-AGE and MD2-TLR4 interaction in HG-stimulated cells. These results show that 1) HG-mediated ROS increases the expression of TLR4, and 2) ROS may not be involved in HG-induced MD2-TLR4 complex formation and MD2-TLR4 activation in a significant manner. These new data have been incorporated and corresponding changes to the text have been made.

New Supplementary Figure S17: Intracellular HG increases the expression of TLR4 via ROS but is not involved in MD2-TLR4 complex formation and TLR4 activation. (A) MD2-TLR4 interaction in mouse primary macrophages (MPM) exposed to HG with or without ROS scavenger N-acetyl cysteine (NAC; 5 mM) pretreatment. Cells were exposed to 33 mM HG for 15 minutes following a 1-hour pretreatment with 5 mM NAC. MD2 was immunoprecipitated (IP) and interaction with TLR4 was determined by immunoblotting (IB). (B) MD2-TLR4 interaction in H9C2 cells. Cells were treated as indicated for Panel A. (C) Mouse primary macrophages were exposed to 33 mM glucose (HG) for 12 hours, with or without 1h pretreatment with NAC (5 mM). mRNA levels of TNF- α and IL-6 were measured by real-time qPCR [mRNA data normalized to β -actin; Mean \pm SEM; n = 3; * p <0.05, ** p <0.01]. (D) TNF- α and IL-6 mRNA levels in H9C2 cells. Cells were treated as indicated for Panel C. (E) TLR4 expression in mouse primary macrophages and H9C2 cells following exposure to HG. Cells were treated as indicated for Panel C [mRNA data normalized to β -actin; Mean \pm SEM; n = 3; * p <0.05, ** p <0.01].

Reviewer #1-3: Convincing data show a role of TLR2 in diabetic Nephropathy see Devaraj et al ATVB, etc - you cannot ignore published data to fit your hypothesis.

Response: Thank you for this comment. It was not our intention to ignore any relevant published study. Our objective was and remains elucidating the role of TLR4 in diabetic cardiomyopathy, as stated in the abstract and introduction while setting the context. Another

reason is that MD2 is not an assistant and necessary protein for TLR2 signaling pathway. We also validated that HG could not induced interaction between MD2 and TLR2 in primary macrophages and H9C2 cells (**Reviewer Figure 1A** below). We do acknowledge that TLR2 may be involved in pathogenic cardiac changes taking place in diabetes. We validated that high level of glucose increase TLR2 expression in macrophages and H9C2 cells (**Reviewer Figure 1B** below). To elucidate the contribution of TLR2 in inflammatory phenotype of cells exposed to HG and AGE, we have performed an additional study. We knocked down the expression of TLR2 in RAW264.7 macrophage line and then exposed the cells to either HG or AGE-BSA. We measured MD2-AGE/TLR4 interaction and induction of inflammatory cytokines. Our results (**Reviewer Figure 1C-1F** below) show that TLR2 knockdown does not reduce the interaction between MD2-AGE and MD2-TLR4 upon HG or AGE-BSA exposure. Furthermore, TLR2 silencing reduced HG-induced TNF- α and IL-6 expression (although, these were not completely prevented) but had no effect on TNF- α and IL-6 expression in response to AGE-BSA. These results show that TLR2 may participate in HG-induced inflammatory factor expression, but AGE-mediated changes are primarily driven through TLR4.

In our honest opinion, including TLR2 data disrupts the information flow and makes the study unfocused. However, we are providing the data to the reviewer with the hope that these new studies address the reviewer's concern of including the potential involvement of TLR2.

Reviewer Figure 1: TLR2 does not mediate AGE-induced inflammatory factor expression in macrophages. (A) MD2 has no interaction with TLR2. Mouse primary macrophages

(MPM) and H9c2 cells were stimulated with or without 33mM HG for 15 min. MD2 was immunoprecipitated (IP) and interaction with TLR2 was determined by immunoblotting (IB). (B) Mouse primary macrophages and H9c2 cells were exposed to 33 mM glucose (HG) for 12 hours. mRNA levels of TLR2 were measured by real-time qPCR [mRNA data normalized to β -actin; Mean \pm SEM; n = 3; *** p <0.001]. (C) RAW264.7 macrophages were transfected with negative control siRNA (NC) or siRNA targeting TLR2 (siTLR2). Cells were then exposed to 33 mM glucose (HG) for 15 minutes. Proteins were isolated, immunoprecipitated (IP) using MD2 antibody. Interaction with AGE, TLR4, and TLR2 was assessed by immunoblotting (IB). (D) siRNA transfected RAW264.7 cells were exposed to 30 μ g/mL AGE-BSA and interaction between MD2-AGE and MD2-TLR4 was probed. (E) RAW264.7 cells were exposed to 33 mM glucose for 12 hours, following transfection with NC or siTLR2. mRNA levels of TNF- α and IL-6 were measured by qPCR [mRNA data normalized to β -actin; Mean \pm SEM; n = 3; * p <0.05, ** p <0.01, and *** p <0.001]. (F) RAW264.7 cells were exposed to 30 μ g/mL AGE-BSA for 12 hours, following transfection with NC or siTLR2. mRNA levels of TNF- α and IL-6 were measured by qPCR [mRNA data normalized to β -actin; Mean \pm SEM; n = 3; ns = not significant, *** p <0.001].

Reviewer #1-4: Diabetes spans many decades what is the need to show an effect of 15 minutes of HG. What is the clinical relevance?

Response: As the reviewer knows, experimental models are used to understand the biology of the disease (mechanisms, points of intervention etc.). We are not aware of any disease model, where the time-course of disease initiation and progression precisely matches the human condition. Key molecular events that take years to develop in humans may only take months in mice and minutes to hours in cultured cells. For example, inflammatory cytokines are observed to be elevated in human diabetes (following years of disease duration), rodent models of diabetes (months of duration), and following exposure of cultured cells to high levels of glucose (within hours of exposure). Despite this disconnect in the time-course, it is well established that diabetes produces a state of chronic inflammation, manifesting as elevated proinflammatory cytokine levels and downstream effects. In addition, although the high concentration of glucose in cultural medium will be decreased with the passage of time, the in vivo hyperglycemia state is continuous in diabetic disease, indicating that the event of MD2/TLR4 activation may continuously happen. Therefore, recapitulation of the key pathogenic readout is perhaps more important in assigning clinical relevance. Based on our studies presented in the manuscript, high levels of circulating glucose in diabetes is anticipated to generate AGE products which engage MD2-TLR4 signaling pathway to induce inflammatory cytokines and lead to cardiac dysfunction. We, to the best of our knowledge and ability, show that disrupting this axis in cultured cells and mouse models of diabetes, protects the cells and heart tissues. Collectively, our cell culture mechanistic studies, examination of tissues in diabetic mice, and analysis of samples from patients with diabetes provides ample support for clinical relevance.

Reviewer #1-5: I am not convinced what is your AGE moiety. You should repeat experiments

with AGE that has MG or CML etc. Otherwise your data is troubling. If MG and CML does not cross react in your assay I still maintain (which you ignored 1-5) that your assay is unacceptable and not validated for mainstream research.

Response: Respectfully, we did not ignore the reviewer's concerns. We, **1)** exposed H9C2 cells to CML-BSA and did not observe interaction between MD2 and TLR4 (Reviewer 1-4; data added to Supplementary Figures), **2)** measured endotoxin levels in our AGE-BSA (Reviewer 1-5; text added), **3)** confirmed the results of MD2-TLR4 interaction and TNF- α /IL-6 induction in the presence of polymyxin B (Reviewer 1-6; data added to Supplementary Figures), and **4)** quantified MG and CML in cultured cells and heart tissues using a commercially available kit or published technique (Reviewer 1-7; data added to Supplementary Figures).

To build on our results (and confirm the concepts), we have performed additional experiments. First, we incubated complete growth media with 33 mM glucose for 15 minutes and measured the levels of AGE-, MGO-, and CML-modified proteins, using commercially available and validated assay kits. Our results show that glucose exposure of serum-containing media rapidly generates AGE-modified serum proteins. However, we did not observe an increase in MGO- or CML-modified proteins (new *Supplementary Figure S22D*).

Then, we exposed mouse primary macrophages to 30 μ g/mL AGE-BSA (PMID: 20133001), MGO-BSA (PMID: 31253329), or CML-BSA (PMID: 20133001) for 15 minutes. We then measured the interaction between MD2-AGE and MD2-TLR4 by immunoprecipitation. We also measured downstream signaling by assessing NF- κ B activation and induction of inflammatory cytokines. Our results show that AGE-BSA increases the interaction between MD2-AGE and MD2-TLR4, and leads to NF- κ B activation (as assessed by reduced levels of inhibitor of κ B) and TNF- α /IL-6 induction. These changes were not noted when cells were exposed to MGO- and CML-modified BSA. We hope that this additional data (added as new *Supplementary Figure S23*) provides support for our conclusions. We do not know why MGO- and CML-modified proteins are not elevated and do not interact with MD2, as stated in our manuscript. We suspect that this may be due to the experimental setup. It is possible that given more time, CML- and MGO-modified proteins may be elevated. However, since MD2-TLR4 complex formation and downstream pathway activation is quick, we have focused on changes that take place with glucose incubation of up to 30 minutes. We also indicated in the discussion section there are technical challenges with identifying the exact binding site of AGE-modified proteins on MD2. However, the results of functional downstream significance are reliable and consistent.

Supplementary Figure S22D: HG incubation in cell-free medium do not produce MGO and CML. Complete growth media containing serum was incubated with 33 mM glucose (HG) for 15 minutes. Levels of AGE-, MGO-, and CML-modified proteins were measured by the methods described in Methods section [n = 3; ns = not significant compared to respective complete media without HG; ***p<0.001 compared to respective complete media without HG].

Supplementary Figure S23: AGE-BSA but not MGO-BSA or CML-BSA activates MD2/TLR4 inflammation. (A) Mouse primary macrophages were exposed to 30 μg/mL AGE-, MGO-, or CML-modified BSA for 15 minutes. Proteins were isolated, immunoprecipitated by MD2 antibody, and interaction between MD2-TLR4-MyD88 were assessed by immunoblotting. Cells in control group were exposed to 30 μg/mL BSA protein. Representative blots were shown from three independent experiments. (B) MD2-TLR4-MyD88 interaction in H9C2 cells. H9C2 cells were treated as indicated for Panel B. (C) Primary macrophages or H9c2 cells

were exposed to 30 $\mu\text{g}/\text{mL}$ AGE-BSA, MGO-BSA, or CML-BSA for 1 hour. Cell lysates were probed for level of inhibitor of κB ($\text{I}\kappa\text{B}\alpha$). GAPDH was used as loading control. Cells in control group were exposed to 30 $\mu\text{g}/\text{mL}$ BSA protein. Representative blots were shown from three independent experiments. (D) Real-time qPCR assay shows the mRNA levels of $\text{TNF-}\alpha$ and IL-6 in primary macrophages exposed to 30 $\mu\text{g}/\text{mL}$ AGE-BSA, MGO-BSA, or CML-BSA for 12 hours [Con = 30 $\mu\text{g}/\text{mL}$ BSA; $n = 9$; $**p < 0.01$; ns = no significance].

Reviewer #1-6: I need to see MD2 in mononuclear Cells as protein in T2DM patients without CMO

Response: Thank you for the suggestion. We obtained 7 blood samples from type 2 diabetic patients without cardiomyopathy. We have measured MD2 protein levels in peripheral blood mononuclear cells and showed the data in **Reviewer Figure 2** below. The results show that the MD2 protein level PBMCs in diabetic subjects without cardiomyopathy is slightly higher than in normal subjects but significantly lower than that in DCM patients. In our honest opinion, including the data from diabetic subjects without cardiomyopathy does not match the data from animals and may make the study unfocused on DCM. We did not show it in the revised Supplementary Figure S29.

Reviewer Figure 2: Western blot analysis of MD2 protein in peripheral blood mononuclear cells isolated from blood samples of nondiabetic individuals and diabetic patients with (DCM) and without cardiomyopathy (D -CM). GAPDH was used as loading control. 3 samples per group ($n = 7$) are shown as representative in immunoblots. Densitometric quantification is shown in the below panel [$n = 7$; ns = not significant, $*p < 0.001$ compared to nondiabetic controls (Con) and D -CM].

Reviewer #1-7: I am unclear what is the relevance of circulating MD2

Response: This is an excellent question. We believe that circulating (soluble; s) MD2 may facilitate AGE-MD2 complex formation in circulation and contribute to TLR4 activation in the heart. We have also shown elevated levels of serum AGE-MD2 complexes in samples obtained from diabetic patients and from experimental models of diabetes.

Although the exact function of sMD2 is not fully defined, studies have shown that complex

formation between sMD2 and LPS increases circulating MD2 protein half-life (PMID: 15175334) and that this complex interacts with TLR4-expressing cells to confer LPS sensitivity (PMID: 11593030). Interestingly, even without interacting with LPS, sMD2 is able to bind cell surface TLR4 and increase its expression (PMID: 18845299). Therefore, the consensus is that sMD2 facilitates innate immune responses, in part through activating TLR4 in MD2-deficient cells. We have included this text in the revised discussion.

Besides the biological relevance, with further investigation, circulating AGE-MD2 level could prove to a blood indicator/ biomarker of diabetic complications in the future. However, this certainly requires further investigation and confirmation.

REVIEWER 2

Reviewer #2-1: In this revised manuscript, the authors have done a thorough job to address all the concerns raised with extensive amount of new data. The manuscript has been significantly improved. I have only one minor comment. The author should list heart weights, and heart weight over body weight ratios in the paper since this is a critical perimeter for cardiac remodeling in diabetic cardiomyopathy.

Response: Thank you very much. We have included data on heart weights and heart weight to body weight ratios in the revised Table 1 (revised data shown below).

Parameter ¹	WT-Con (n=6) ²	WT-STZ (n=7) ²	MD2KO-Con(n=6) ²	MD2KO-STZ(n=7) ²
HW(mg)	125.1 ± 3.28	117.8 ± 1.42	131.0 ± 4.55	116.0 ± 2.92
BW(g)	29.68 ± 0.95	25.30 ± 0.38	32.17 ± 1.70	27.07 ± 0.48
HW/BW(mg/g)	4.218 ± 0.065	4.660 ± 0.031**	4.093 ± 0.082	4.281 ± 0.075#

Table 1 (addition). Data shown in red font has been added to the revised table [HW = heart weight, BW = body weight, HW/BW = heart weight to body weight ratio; **p<0.01 compared to WT-Con; #p<0.05 compared to MD2KO-Con].

REVIEWER 3

Reviewer #3-1: The authors have taken great effort to revise their manuscript on the role of MD2 in the development of inflammatory diabetic cardiomyopathy and have improved their work. While the presented additional mechanistic data are compelling, I have to admit that I am still not convinced that there is really a clinical relevance to the model used (see my previous Q1). The given answer that “other pathological indices of cardiac damage are also taken into consideration” does not alter the fact that an ejection fraction of the left ventricle above 70% is far from any relevant impairment that could lead to heart failure (symptoms). If the argument is taken that diastolic heart function might play a role, then stringent further analysis would be required (see e.g. doi: 10.1038/s41586-019-1100-z).

Response: Thank you for this comment. As mentioned, our results are consistent with studies in both streptozotocin-induced diabetic mice and rats showing cardiac dysfunction with

preserved ejection fraction. As suggested by the reviewer, however, we have included new data indicative of diastolic dysfunction in the model system used. Specifically, we have included peak velocities during late diastole (A) to calculate E/A ratios (early diastole (E) was already included in the previous version of the manuscript), and the isovolumic relaxation time (IVRT). Both measurements show increased levels in streptozotocin-induced diabetic mice at week 16, indicating diastolic dysfunction. We hope that with this revision, we can address the reviewer’s concern.

Parameter ¹ ↕	WT-Con (n=6) ² ↕	WT-STZ (n=7) ² ↕	MD2KO-Con (n=6) ² ↕	MD2KO-STZ (n=7) ² ↕
A (m/s) [↕]	0.531 ± 0.02 [↕]	0.322 ± 0.06 [↕]	0.632 ± 0.06 [↕]	0.628 ± 0.08 [↕]
E/A [↕]	0.842 ± 0.13 [↕]	1.615 ± 0.20 ^{**} ↕	1.002 ± 0.09 [↕]	1.033 ± 0.15 [#] ↕
IVRT(ms) [↕]	12.1 ± 1.38 [↕]	21.14 ± 1.86 [*] ↕	13.8 ± 2.12 [↕]	16.3 ± 2.22 [#] ↕

Table 1 (addition): Data shown in red font has been added to the revised table [A = peak velocity during late diastole; E/A ratio = ratio of peak velocity blood flow from early diastole to late diastole; IVRT = Isovolumic relaxation time; *p<0.05 and **p<0.01 compared to WT-Con; #p<0.05 compared to MD2KO-Con].

Reviewer #3-2: Another aspect that is irritating is the response to my previous Q8 that “M-Mode/B-Mode ... traces were not obtained”. This appears to be rather odd since these traces/images had to be acquired for the analysis in the first place.

Response: We apologize for the inadequate response. The study and analysis was performed off-site at the Affiliated Second Hospital of the Wenzhou Medical University. We have been successful in locating the original equipment and backed-up data. The images are included in the new **Supplementary Figure S3**. Again, we are apologetic and should have made a more concerted effort to include this data in the first revision.

Supplementary Figure S3: Representative left ventricular M-mode echocardiographic tracings weeks [WT-Con = nondiabetic controls, n=6; WT-STZ = STZ-induced diabetic mice, n=7; MD2KO-Con = nondiabetic MD2-/- mice, n=6; MD2KO-STZ, STZ-induced diabetic MD2-/- mice, n=7].

REVIEWERS' COMMENTS:

Reviewer #2 (Remarks to the Author):

Well done.

Reviewer #3 (Remarks to the Author):

Following an insufficient first round of revisions, the authors now adequately address the raised concerns. While the general limitations of the model used remain, I have no further specific comments.

Response to reviews

Manuscript: MD2 activation by direct AGE interaction drives inflammatory diabetic cardiomyopathy.

REVIEWERS' COMMENTS:

Reviewer 2, comment 1: Well done.

Response: We thank you for reviewing our manuscript again.

Reviewer 3, comment 1: Following an insufficient first round of revisions, the authors now adequately address the raised concerns. While the general limitations of the model used remain, I have no further specific comments.

Response: Thank you again for reviewing our work. We have included a statement on the limitations of the models used.